# Instructed knowledge shapes feedback-driven aversive learning in striatum and orbitofrontal cortex, but not the amygdala

Lauren Y Atlas[1,2]*, Bradley B Doll[3,4], Jian Li[5,6], Nathaniel D Daw[7], Elizabeth A Phelps[3,8,9]*

[1]National Center for Complementary and Integrative Health, National Institutes of Health, Bethesda, United States; [2]National Institute on Drug Abuse, National Institutes of Health, Bayview, United States; [3]Center for Neural Sciences, New York University, New York, United States; [4]Department of Psychology, Columbia University, New York, United States; [5]Department of Psychology, Beijing Key Laboratory of Behavior and Mental Health, Peking University, Beijing, China; [6]PKU-IDG/McGovern Institute for Brain Research, Peking University, Beijing, China; [7]Department of Psychology, Princeton Neuroscience Institute, Princeton University, Princeton, United States; [8]Department of Psychology, New York University, New York, United States; [9]Nathan Kline Institute, Orangeburg, United States

*For correspondence: lauren.
atlas@nih.gov (LYA); liz.phelps@
nyu.edu (EAP)

Competing interests: The authors declare that no competing interests exist.

**Abstract** Socially-conveyed rules and instructions strongly shape expectations and emotions. Yet most neuroscientific studies of learning consider reinforcement history alone, irrespective of knowledge acquired through other means. We examined fear conditioning and reversal in humans to test whether instructed knowledge modulates the neural mechanisms of feedback-driven learning. One group was informed about contingencies and reversals. A second group learned only from reinforcement. We combined quantitative models with functional magnetic resonance imaging and found that instructions induced dissociations in the neural systems of aversive learning. Responses in striatum and orbitofrontal cortex updated with instructions and correlated with prefrontal responses to instructions. Amygdala responses were influenced by reinforcement similarly in both groups and did not update with instructions. Results extend work on instructed reward learning and reveal novel dissociations that have not been observed with punishments or rewards. Findings support theories of specialized threat-detection and may have implications for fear maintenance in anxiety.

## Introduction

Neuroscientists have built a rich understanding of the brain systems that govern learning from punishments and rewards, which seem to be largely conserved across species (*Balleine and O'Doherty, 2010*; *Haber and Knutson, 2010*). Yet humans possess a distinctive capacity to learn from socially-conveyed rules and instructions, which is an important example of a broader capacity, shared by many species, of learning from events other than simple reinforcers (*Tolman, 1949*). Most of us do not need to get burnt to avoid putting our hands on a hot stove: A verbal warning serves as a sufficient threat. How does instructed knowledge influence our subsequent responses to ongoing threats in the environment?

**eLife digest** Around the start of the twentieth century, Pavlov discovered that dogs salivate upon hearing a bell that has previously signaled that food is available. This phenomenon, in which a neutral stimulus (the bell) becomes associated with a particular outcome (such as food), is known as classical conditioning. The network of brain regions that supports this process – which includes the striatum, the amygdala and the prefrontal cortex – seems to work in a similar way across most animal species, including humans.

However, humans don't learn only through experience or trial-and-error. We do not need to burn our hands to learn not to touch a hot stove: a verbal warning from others is usually sufficient. Experiments have shown that giving people verbal instructions on how to obtain rewards alters the activity of the striatum and prefrontal cortex. That is, the instructions interact with the circuit that also supports learning through experience. But is this the case for learning how to avoid punishments? That process depends largely on the amygdala, and it is possible that systems designed to detect threats may be less sensitive to verbal warnings.

To address this question, Atlas et al. taught people to associate one image with a mild electric shock, and another with the absence of a shock. After a number of trials, the relationships were reversed so that the previously neutral picture now predicted a shock and vice versa. Telling the participants about the reversal in advance triggered changes in the activity of the striatum and part of the prefrontal cortex. By contrast, such warnings had no effect on the amygdala. Instead, the activity of the amygdala changed only after the volunteers had experienced for themselves the new relationship between the pictures and the shocks.

A key next step is to find out whether this distinction between the two types of learning signals (those that can be updated by instructions and those that cannot) is specific to humans. While the current study relied upon language, there are other methods that could be used to explore this issue in animals. Furthermore, knowing that the human brain has a specialized threat detection system that is less sensitive to instructions could help us to understand and treat anxiety disorders. Atlas et al. hope to test this possibility directly in the future.

We tested whether dynamic feedback-driven aversive learning is modulated when individuals are informed about contingencies in the environment. It is unknown whether instructions integrate with the systems that support error-driven learning, as most reinforcement learning models fail to account for the potential influence of explicit information, despite the fact that verbal information shapes responses across nearly all domains, including aversive learning (*Wilson, 1968*; *Phelps et al., 2001*; *Costa et al., 2015*; *Mertens and De Houwer, 2016*). We sought to characterize computationally the contribution of instructed knowledge to dynamic aversive learning in humans. Our aim was to determine whether instructed knowledge influences the neural mechanisms of aversive learning, or whether separate neural systems process feedback-driven and instructed knowledge.

Recent studies in the appetitive domain suggest that instructions about rewarding outcomes modulate learning-related responses in the striatum (*Doll et al., 2009*; *2011*; *Li et al., 2011a*) and ventromedial prefrontal cortex (*Li et al., 2011a*) and that this modulation might depend on the prefrontal cortex (*Doll et al., 2009*; *2011*; *Li et al., 2011a*). In the aversive domain, such interactions, by which instructed knowledge might help to overcome learned expectations of threat, are of particular importance due to their relevance for anxiety and post-traumatic stress disorder. However, no studies have tested whether instructions have the same effects on dynamic aversive learning, which is known to depend on the amygdala (*Maren, 2001*) but also involves the striatum (*Seymour et al., 2004*; *2005*; *Delgado et al., 2008b*) and ventromedial/orbitofrontal cortex (VMPFC/OFC) (*Phelps et al., 2004*; *Kalisch et al., 2006*; *Schiller et al., 2008*). Instructions might modulate learning in the amygdala as well as the striatum and VMPFC/OFC, or amygdala responses might be insensitive to cognitive instruction (*Ohman and Mineka, 2001*), as suggested by theories of automatic threat detection in the amygdala (*Ohman, 2005*).

All participants performed a Pavlovian aversive learning task in which one image, the Original conditioned stimulus (CS+), was paired with mild electric shock (the unconditioned stimulus [US]) on

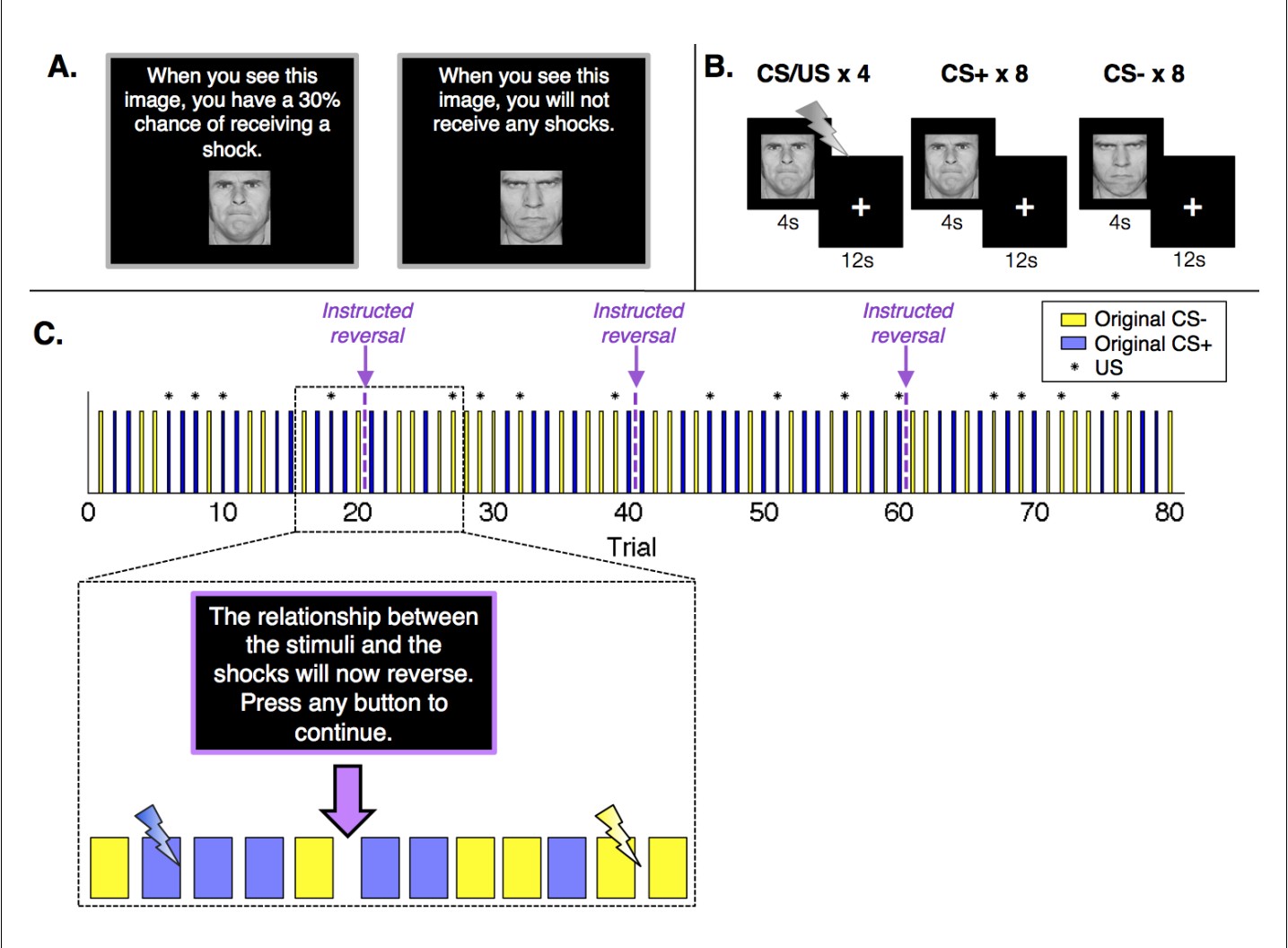

**Figure 1.** Experimental design. (**A**) Prior to the conditioning phase of the experiment, participants in the Instructed Group saw each image and were informed about initial probabilities. Participants in the Uninstructed Group also saw the images prior to the experiment, but were not told about contingencies. (**B**) Participants in both groups underwent a Pavlovian fear conditioning task with serial reversals. There were three reversals across the duration of the task, leading to four continuous blocks of twenty trials. In each block, one image (the conditioned stimulus, or CS+) was paired with a shock (the unconditioned stimulus, or US) 30% of the time, leading to 4 reinforced trials and 8 unreinforced trials, whereas a second image (the CS-) was never paired with a shock. Images were presented for 4 s, followed by a 12-second inter-stimulus interval. (**C**) Upon each reversal, the Instructed Group was informed that contingencies had reversed. Button presses were included to ensure participants were paying attention to the instructions but had no effect on the task itself or task timing. Instructions were always immediately followed by at least two unreinforced presentations of each CS before the new CS+ was paired with a shock. The figure presents one of two pseudorandom trial orders used during the experiment (see Materials and methods).

30% of trials, and a second image, the Original CS-, was not paired with shock. Contingencies reversed three times. Participants assigned to an Instructed Group were informed about initial contingencies and instructed upon reversal (*Figure 1*), whereas participants in an Uninstructed Group learned through reinforcement alone. Models were fit to skin conductance responses (SCRs), a traditional measure of the conditioned fear response in humans. We combined quantitative modeling of behavior with functional magnetic resonance imaging (fMRI) to examine how instructions influence brain responses and skin conductance responses (SCRs) during fear conditioning. We evaluated quantitative learning models, fit to SCR, and confirmed model conclusions with task-based fMRI analyses. We focused on responses in the amygdala, striatum, and VMPFC/OFC. We hypothesized that instructions about contingencies would modify learning-related signals and brain responses

**Table 1.** Group differences in differential SCR[a].

| Analysis | Model | Intercept | Stimulus (Original CS+ > Original CS-) | Reversal effect (Original contingencies vs. Reversed contingencies) | Stimulus x Reversal Interaction (Current CS+ > Current CS-) | Time |
|---|---|---|---|---|---|---|
| All participants (n = 68) | Within-subjects effects, controlling for group (first level) | β = 0.22<br>t = 12.37<br>p <0.0001 | n.s. (p = 0.06) | n.s. (p = 0.445) | β = 0.04<br>t = 5.82<br>p<0.0001 | β = -.07<br>t = 10.59<br>p<0.0001 |
| | Effect of group (second level) | n.s. (p = 0.11) | n.s. (p = 0.15) | n.s. (p = 1.0) | β = 0.03<br>t = 3.98<br>p = 0.0002 | n.s. (p = 0.09) |
| Learners only (n = 40) | Within-subjects effects, controlling for group (first level) | β = 0.27<br>t = 11.62<br>p<0.0001 | β = 0.01<br>t = 2.79<br>p = 0.0082 | n.s. (p = 0.95) | β = 0.06<br>t = 6.5<br>p<0.0001 | β = -.08<br>t = -11.18<br>p<0.0001 |
| | Effect of group (second level) | n.s. (p = 0.13) | n.s. (p = 0.20) | n.s.(p = 0.35) | β = 0.04<br>t = 3.67<br>p = 0.0007 | β = -.02<br>t = -2.53<br>p = 0.0157 |
| Instructed Group learners (n = 20) | Entire task | β = 0.30<br>t = 8.93<br>p<0.0001 | n.s. (p = 0.30) | n.s. (p = 0.55) | β = 0.10<br>t = 5.35<br>p<0.0001 | β = -0.09<br>t = -9.26<br>p<0.0001 |
| | Entire task, second half of each run | β = 0.27<br>t = 7.82<br>p<0.0001 | β = 0.02<br>t = 2.15<br>p = 0.0314 | β = 0.04<br>t = 4.18<br>p<0.0001 | β = 0.12<br>t = 6.19<br>p<0.0001 | n.s. (p = 0.69) |
| | Trials following the first reversal | β = 0.28<br>t = 6.89<br>p<0.0001 | n.s. (p = 0.65) | β = -0.16<br>t = -2.36<br>p = 0.0184 | β = 0.09<br>t = 4.19<br>p<0.0001 | β = -0.07<br>t = -5.05<br>p<0.0001 |
| | Trials following the first reversal, second half of each run | β = 0.22<br>t = 5.67<br>p<0.0001 | n.s.(p = 0.81) | n.s. (p = 0.30) | β = 0.09<br>t = 4.49<br>p<0.0001 | n.s. (p = 0.09) |

*Table 1 continued on next page*

*Table 1 continued*

| Analysis | Model | Intercept | Stimulus (Original CS+ > Original CS-) | Reversal effect (Original contingencies vs. Reversed contingencies) | Stimulus x Reversal Interaction (Current CS+ > Current CS-) | Time |
|---|---|---|---|---|---|---|
| Uninstructed Group learners (n=20) | Entire task | β = 0.23 t = 7.30 p<0.0001 | β = 0.01 t = 2.09 p = 0.0368 | n.s. (p = 0.54) | β = 0.03 t = 4.58 p<0.0001 | β = -.06 t = -6.10 p<0.0001 |
| | Entire task, second half of each run | β = 0.22 t = 6.86 p<0.0001 | β = 0.02 t = 3.18 p = 0.0015 | β = 0.02 t = 2.44 p = 0.0149 | β = 0.04 t = 5.45 p<0.0001 | n.s.(p = 0.678) |
| | Trials following the first reversal | β = 0.21 t = 5.28 p<0.0001 | n.s. (p = 0.58) | n.s. (p = 0.49) | β = 0.01 t = 1.67 p = 0.0967 | β = -.04 t = -2.49 p = 0.0127 |
| | Trials following the first reversal, second half of each run | β = 0.19 t = 5.83 p<0.0001 | n.s. (p = 0.65) | n.s. (p = 0.21) | β = 0.02 t = 2.04 p = 0.0413 | β = 0.03 t = 2.88 p = 0.004 |

[a]This table presents results of linear mixed models that included normalized skin conductance response (SCR) as a dependent measure. In the Instructed Group, contingencies and reversals are coded relative to instructed reversal. In the Uninstructed Group, contingencies and reversals are coded relative to reinforcement (i.e. reversals occur when the previous CS- is paired with a shock). Within groups, we analyzed SCRs across the entire task as well as following the first reversal (without the acquisition phase). We also examined responses within the second half of each run, as well as across all trials (including trials that immediately followed reversals).

during fear conditioning, and that, as in the appetitive domain, this modulation would involve the prefrontal cortex.

## Results

### Subjective ratings

Upon completion of the experiment, participants reported the number of perceived contingency reversals and retrospectively rated shock expectancy and affect in response to each image. Participants recognized that three reversals had occurred (*M* = 3.22, *SD* = 0.79), and there was no group difference in the estimated number of reversals (p>0.1). Two-way ANOVAs revealed a main effect of Group on reported affect (F(1,68) = 8.57, p<0.01), such that the Uninstructed Group reported less positive affect for both stimuli. There were no Group differences in shock expectancy ratings, nor were there any main effects of Stimulus (Original CS+ vs Original CS-) or Group x Stimulus interactions on either outcome measure.

### Instructions influence skin conductance responses

We tested whether participants showed differential SCRs to unreinforced CS presentations during fear acquisition (i.e. larger responses to the Original CS+ than Original CS-), and whether responses were modulated after participants were instructed that contingencies had reversed (Instructed Group) or after they received a shock paired with the new CS+/ previous CS- (Uninstructed Group). As reported in *Table 1*, both groups showed differential responses that reversed in response to contingency changes throughout the task (CS+ > CS-; ß = 0.04, t = 5.82, p<0.0001). Differential responses were larger in the Instructed Group than the Uninstructed Group (ß = 0.03, t = 3.98, p=0.0002), and SCRs habituated over time (ß = -0.07, t = 10.59, p<0.0001).

These results reflect group-level effects and group differences across all participants. However, a subset of participants did not show differential SCRs prior to the first reversal. As our primary research question concerns the effects of instructions on the neural systems of aversive reversal learning, the strongest tests are in those individuals who learn contingencies prior to the first reversal. Given this, we restricted our analyses to 'learners.' As described in Materials and methods, we defined learners as those individuals who showed greater SCR to the CS+ relative to the CS- in late acquisition (the second half of the first run; 20/30 Instructed Group participants, 20/38 Uninstructed Group participants). In this subset of learners, the main effects and interactions on SCR reported above remained significant and increased in magnitude (*Figure 2A,C*; *Table 1*). Importantly, both the Instructed and Uninstructed Groups showed SCRs that were responsive to changing contingencies when we examined each group separately, and when we restricted analyses to trials that followed the first reversal (*Table 1*), suggesting that effects were not driven entirely by initial learning. The Uninstructed Group showed significant reversals when we analyzed trials from the second half of each block in this post-acquisition analysis, and marginal effects when we included all post-acquisition trials (see *Table 1*). This is expected given the low reinforcement rate (33%) that we used in this study, and is consistent with somewhat slow learning. The quantitative models and fMRI analyses reported below focus on the 20 learners in each group. Combined fMRI analyses that include data from the full sample are entirely consistent with these findings and are reported in figure supplements and source data.

We also tested whether instructions immediately update autonomic responses by examining instructed reversals in Instructed Group learners. Each reversal featured a delay between instruction delivery and reinforcement of instructions (*Figure 1C*): Following instructions, each CS was presented without reinforcement at least twice before the previous CS- was paired with a US. We compared responses during these post-instruction, pre-reinforcement windows with an equivalent number of trials prior to each instruction (e.g. the last two CS+ and CS- trials in each phase). We found a significant effect of instructions on differential responding (ß = 0.04, t(19) = 3.50, p= 0.0024), such that SCR responses reversed immediately after each instruction, before any actual reinforcement was delivered.

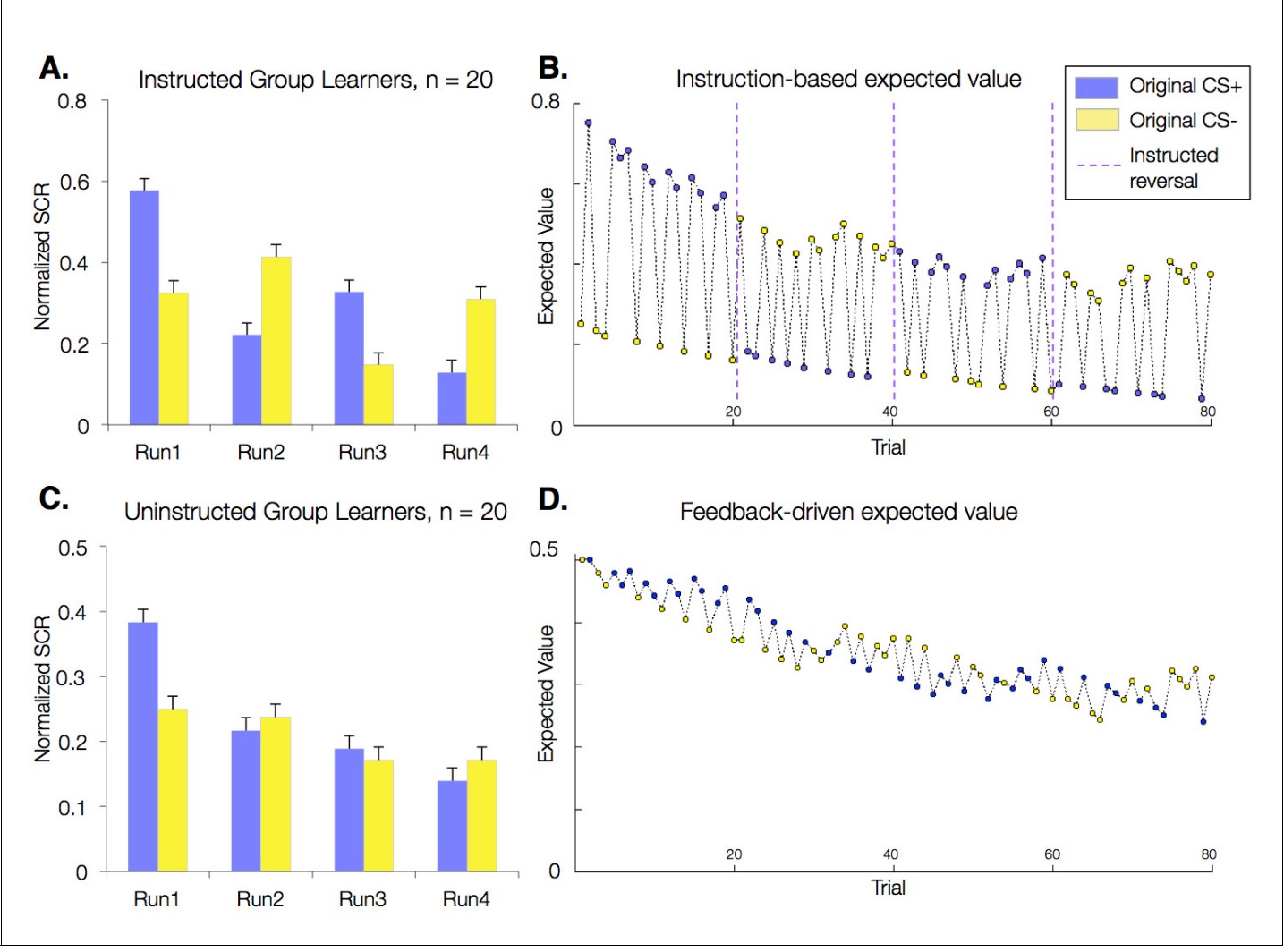

**Figure 2.** Effects of instructions on skin conductance responses (SCR) and aversive learning. Mean normalized skin conductance responses (SCRs) as a function of group and condition. Both groups showed significant reversals of SCR responses throughout the task (p<0.001; *Table 1*), and effects were larger in the Instructed Group (see *Table 1*). Error bars reflect within-subjects error. (A) Mean SCR in the Instructed Group as a function of original contingencies. Runs are defined relative to the delivery of instructions. (B) Dynamics of expected value based on fits of our modified Rescorla-Wagner model, fit to SCR in the Instructed Group. Fitted model parameters were consistent with SCR reversing almost entirely in response to instructions (ρ = 0.943). This timecourse was used in fMRI analyses to isolate regions involved in instruction-based learning. (C) Mean SCR in the Uninstructed Group as a function of original contingencies. A new run is defined when the previous CS- is paired with a shock. (D) Dynamics of expected value based on the model fit to SCR from the Uninstructed Group. This timecourse was used in fMRI analyses to isolate regions involved in feedback-driven learning in both groups.

## Feedback-driven learning is modulated by instructions

We were interested in understanding how instructions shape error-driven learning and the development of expectations. To this end, we focused on the computation of expected value (EV) and used a simple Rescorla-Wagner model modified to capture flexible effects of instructions and test for dissociations. Our quantitative models assume that SCR at cue onset reflects EV (see Materials and methods). Shocks were incorporated as reinforcements, and thus positive EV corresponds to an expectation for a shock. Consistent with standard Rescorla-Wagner models, EV updates in response to prediction error (PE), and the speed of updating depends on learning rate (α). We focus on correlations with EV in this manuscript, due to concerns of algebraic collinearity when EV, shock, and PE are included in the same model.

**Table 2.** Quantitative model of instructed learning: Rescorla-Wagner modulated with instruction parameter (ρ).

| Group | Analysis type | α | ρ | Deviance |
|---|---|---|---|---|
| Instructed Group learners (n = 20) | Across-subjects analysis | 0.061 | 0.943 | 60.83 |
| | Within-subjects analysis | $M = 0.07, SD = 0.09$ | $M = 0.69, SD = 0.32$ | $M = 2.73, SD = 0.88$ |
| Uninstructed Group learners (n = 20) | Across-subjects analysis | 0.042 | 0 | 40.86 |
| | Within-subjects analysis | $M = 0.11, SD = 0.22$ | $M = 0.10, SD = 0.25$ | $M = 1.98, SD = 0.93$ |

Models fit to the Uninstructed Group's SCRs isolate learning-related processes that respond to reinforcement history alone (i.e. feedback-driven learning), since this group was not informed about cue contingencies or reversals. Models fit to the Instructed Group's SCRs, however, should capture the immediate effects of instruction reported above. To acknowledge this flexible effect of instructions, we modified the standard Rescorla-Wagner model. We introduced an instructed reversal parameter, $\rho$, which determines the extent to which EV reverses upon instruction (see Materials and methods). If ρ = 1, the EVs of the two CSs are swapped completely when instructions are delivered, whereas if ρ = 0, each CS maintains its current EV and the model reduces to a standard experiential Rescorla-Wagner model. The best-fitting parameters when fit across Instructed Group subjects revealed that EV reversed almost completely at the time of instructions in the Instructed Group (ρ = 0.943; *Table 2*), suggesting that instructions immediately influence EV/SCR, as illustrated in *Figure 2B*. When we fit the same model to SCRs from the Uninstructed Group (for whom the additional effect should not be observed since no instructions were given) estimates were indeed consistent with associations not reversing at the time when the instructions would have been delivered (ρ = 0.0; *Table 2*). The resulting time course, which captures slower reversals of EV based on purely feedback-driven learning, is depicted in *Figure 2D*. We also fit the model to individual participants in both groups (see Materials and methods) and found that instructed reversal parameters (i.e. ρ) differed significantly as a function of Group (*Instructed Group > Uninstructed Group*, t(38) = 6.53, p<0.0001; *Table 2*).

## Neural correlates of feedback-driven and instructed aversive learning

Our goal was to determine whether and how the neural systems that support feedback-driven aversive learning are modulated by instructions. Thus our computational neuroimaging analyses proceeded in two stages, guided by the quantitative models reported above. First, we examined each group separately using the model fit to behavior in that group. Thus we used the model depicted in *Figure 2D* to isolate neural correlates of feedback-driven learning in the Uninstructed Group, and the model depicted in *Figure 2B* to isolate neural correlates of instruction-based learning in the Instructed Group. Next, we directly compared the two sources of learning in the Instructed Group, since these participants were exposed to both forms of feedback (i.e. instructions about contingencies and reversals, as well as experiential learning from reinforcement between each reversal). In each analysis, we focused on results in amygdala, striatum, and VMPFC/OFC (see *Figure 3—figure supplement 3*) to determine whether these *a priori* regions of interest (ROIs) were sensitive to feedback-driven learning and/or whether they updated with instructions. Finally, we tested the conclusions from quantitative models with task-based analyses that relied strictly on our experimental design, thus eliminating the influence of assumptions derived from our models.

## Neural correlates of feedback-driven aversive learning

We first focused on the neural correlates of experiential learning by examining responses in the Uninstructed Group. Regressors were based on the best-fitting parameters from the model fit to the Uninstructed Group, thus isolating feedback-driven EV (*Figure 2D*; see Materials and methods). ROI-based analyses within the Uninstructed Group revealed a main effect of Region (F(2,40) = 5.13, p=0.011). Post-hoc t-tests revealed that this was driven by positive correlations between feedback-driven EV and responses in the amygdala (bilateral: t(1,19) = 4.21, p=0.0005; Left: t(1, 19) = 3.94, p=0.001; Right: t(1,19): = 3.72, p=0.001) and striatum (bilateral: t(1,19) = 2.31, p=0.0017; left: t(1,

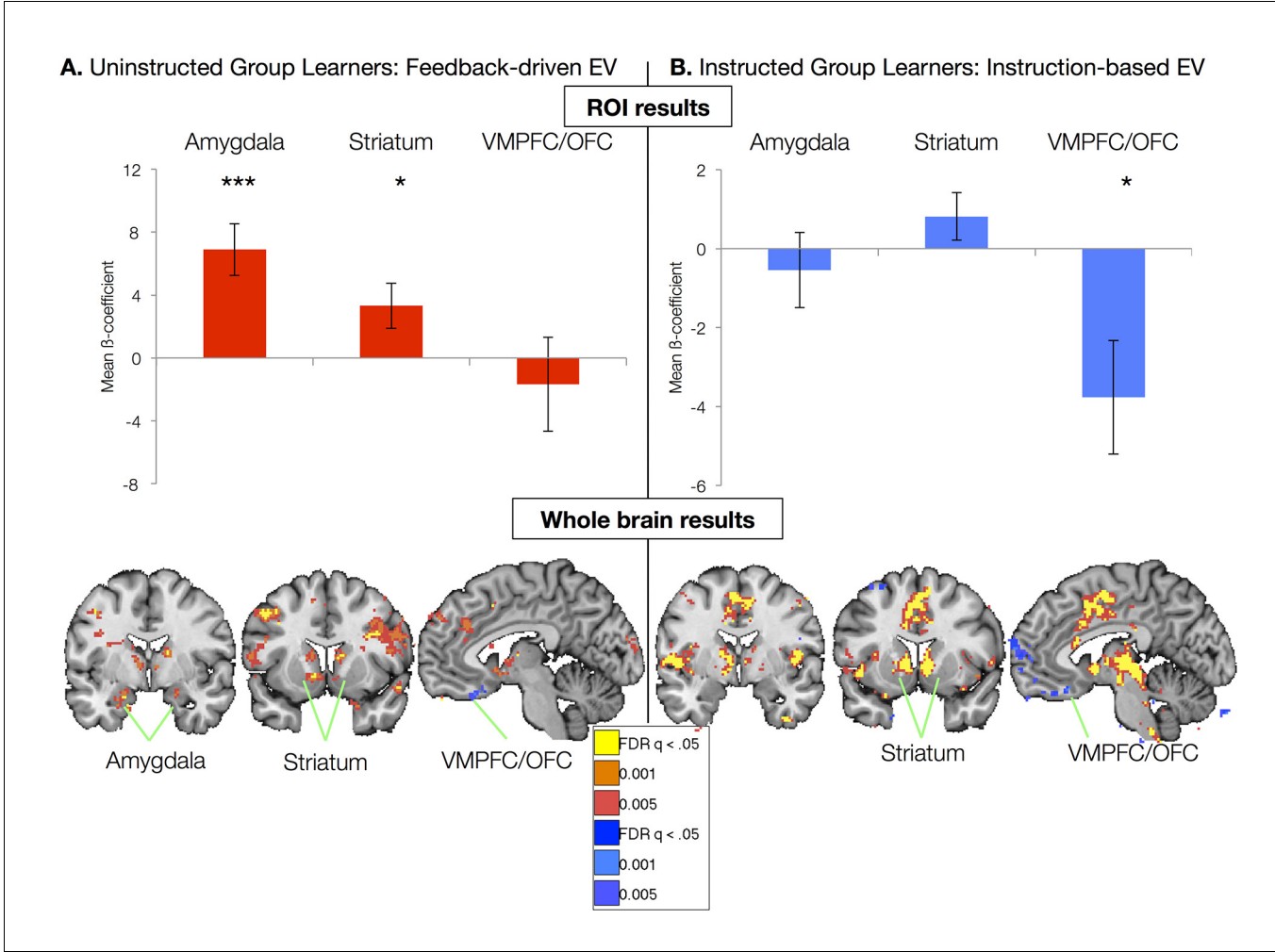

**Figure 3.** Neural correlates of expected value. (**A**) Neural correlates of feedback-driven expected value (EV) were isolated by examining correlations between the timecourse depicted in **Figure 2D** and brain activation in response to cue onset in Uninstructed Group learners (n = 20). *Top:* ROI-based analyses (see **Figure 3—figure supplement 3**) revealed significant correlations with feedback-driven EV in the amydala and striatum. Error bars reflect standard error of the mean; ***p<0.001; *p<0.05. *Bottom:* Voxel-wise FDR-corrected analyses confirmed ROI-based results and revealed additional correlations in the VMPFC/OFC, as well as other regions (see **Figure 3—figure supplement 1**, **Figure 3—figure supplement 1—source data 1**, **2**). (**B**) Neural correlates of instruction-based EV were isolated by examining correlations between the timecourse depicted in **Figure 2B** and brain activation in response to cue onset in Instructed Group learners (n = 20). *Top:* ROI-based analyses revealed a significant negative correlation with instruction-based EV in the VMPFC/OFC. *Bottom:* Voxel-wise analyses confirmed these results and revealed strong positive correlations in the bilateral striatum, as well as the dACC, insula, and other regions (see **Figure 3—figure supplement 2** and **Figure 3—figure supplement 2—source data 1** and **2**). We did not observe any correlations between amygdala activation and instruction-based EV.

The following source data and figure supplements are available for figure 3:

**Figure supplement 1.** Feedback-driven EV in the Uninstructed Group.

**Figure supplement 1—source data 1.** Neural correlates of feedback-driven expected value (EV): Uninstructed Group Learners (n = 20).

**Figure supplement 1—source data 2.** Neural correlates of feedback-driven EV: Entire Uninstructed Group (n = 38).

**Figure supplement 2.** Instruction-based EV in the Instructed Group.

**Figure supplement 2—source data 1.** Neural correlates of instruction-based EV: Instructed Group Learners (n = 20).

**Figure supplement 2—source data 2.** Neural correlates of instruction-based EV: Entire Instructed Group (n = 30).

*Figure 3 continued on next page*

*Figure 3 continued*

**Figure supplement 3.** Regions of interest.

19) = 2.58, p=p =0.018; right: p=p =0.063). Voxel-wise FDR-corrected results confirmed ROI-based findings, and isolated additional correlations in the VMPFC/mOFC (see *Figure 3A*, *Figure 3—figure supplement 1*, and *Figure 3—figure supplement 1—source data 1* and *2*). The striatum and amygdala both showed positive correlations with EV, associated with increased activation for the stimulus currently predicting an aversive outcome. The VMPFC/mOFC showed negative correlations with EV, consistent with prior work showing increased VMPFC/OFC responses to conditioned stimuli predicting safe, relative to aversive, outcomes (*Schiller et al., 2008*). Additional regions that correlated with feedback-driven EV are reported in *Figure 3—figure supplement 1* and associated source data.

## Instructions shape responses in VMPFC/OFC and striatum

We used parameters from the best-fitting instructed learning model to isolate the neural correlates of aversive learning that updates expected value on the basis of instructions in the Instructed Group. In this model, EV updates immediately with instructions (*Figure 2b*), consistent with the SCRs measured in the Instructed Group. ROI-based ANOVAs revealed a significant effect of Region (F(2,40) = 6.49, p=0.0038), driven by significant negative correlations with EV in the VMPFC/OFC ROI (t(1,19) = -2.61, p=0.0173; see *Figure 3B*). Voxelwise FDR-corrected results also revealed robust activation in the bilateral caudate, which showed positive correlations with instructed EV (see *Figure 3B*, *Figure 3—figure supplement 2*, *Figure 3—figure supplement 2—source data 1*, and *Figure 3—figure supplement 2—source data 2*). Additional regions that tracked instructed EV in whole brain analyses and results across the entire Instructed Group are reported in *Figure 3—figure supplements 2*, *3* and associated source data.

## Feedback-driven versus instruction-based learning within the Instructed Group

The preceding results, from separate groups using EV signals driven by instructions or feedback, suggest that responses in the amygdala, striatum, and VMPFC/OFC were driven by reinforcement in the Uninstructed Group, while only VMPFC/OFC and striatal responses were sensitive to instructions the Instructed Group. To test for formal dissociations, we directly compared the neural correlates of instructed and feedback-driven aversive learning in Instructed Group participants, who were exposed to both instructions and experiential learning. This within-subjects analysis ensures that potential dissociations indicated above are driven by differences in the computational sources of neural activation, rather than differences in performance between the groups. To isolate brain responses that were sensitive only to reinforcement (despite the presence of instructions about contingencies and reversals), we used the feedback-driven EV regressor generated from the model fit to the Uninstructed Group's behavior, which takes advantage of the fact that trial sequences were identical for both groups. The instruction-driven EV regressor was generated from the model fit to the Instructed Group's behavior and reported above.

We included both EV regressors in a within-subjects voxel-wise analysis in the Instructed Group. Between reversals, experiential learning would be somewhat correlated across models. Thus, to remove shared variance, we did not orthogonalize regressors in this analysis (see Materials and Methods). A contrast across the two EV regressors formally tests whether each voxel is more related to feedback-driven or instruction-driven EV. Results from this contrast are presented in *Figure 4*. Voxelwise FDR-corrected analyses revealed that the bilateral amygdala was preferentially correlated with feedback-driven EV, while the right caudate and left putamen showed preferential correlations with instruction-based EV (see *Figure 4B*, *Figure 4—figure supplement 1*, and *Figure 4—figure supplement 1—source data 1*). ROI-based analyses confirmed voxelwise results, with a main effect of Region (F(2,40) = 3.76, p=0.0324; *Figure 4A*), driven by amygdala correlations with feedback-driven EV but not instructed EV (t(1,19) = 2.57, p=0.0189), although striatal differences between models were not significant when averaged across the

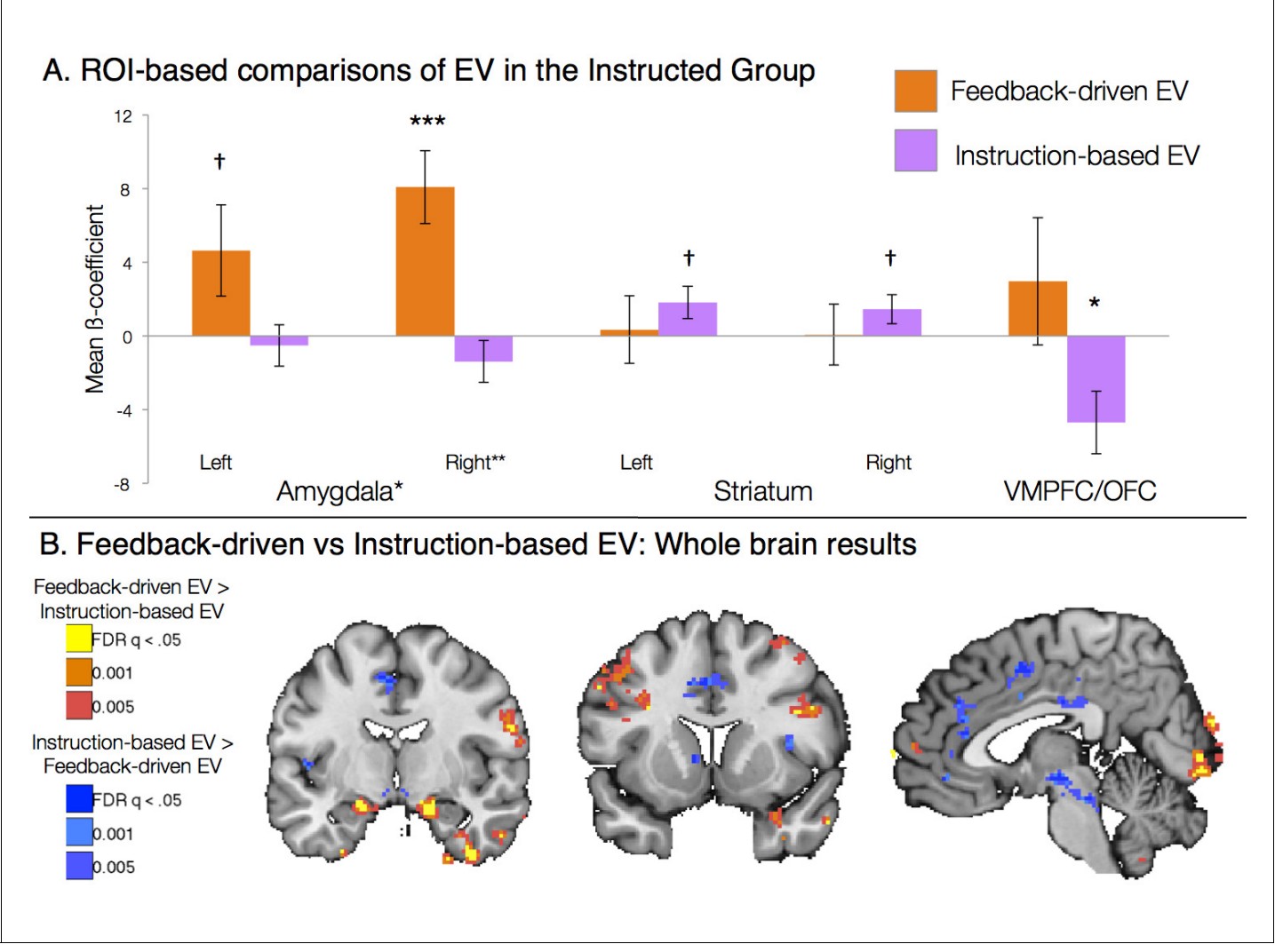

**Figure 4.** Dissociable effects of instructed and feedback-driven learning in the Instructed Group. (**A**) ROI-based effects of feedback-driven and instruction based EV signaling within Instructed Group learners from the model including both signals (see Materials and Methods). Direct model comparisons within Instructed Group learners revealed a significant effect of Model in the amygdala (p<0.05). VMPFC differences were marginal within learners (p = 0.11) and were significant when all Instructed Group participants were included in analyses (p<0.05). Error bars reflect standard error of the mean. ***p<0.001; *p<0.05; †p<0.10. (**B**) Voxelwise direct comparison between feedback-driven and instruction based EV signaling within the Instructed Group. Regions in warm colors, including bilateral amygdala (left), showed preferential correlations with feedback-driven EV. Regions in cool colors, including left caudate (middle), dorsal anterior cingulate and medial prefrontal cortex (right), showed higher correlations with instruction-based EV. Additional regions that showed significant differences as a function of model are presented in *Figure 4—figure supplement 1*, *Figure 4—figure supplement 1—source data 1* and *2*.

The following source data and figure supplements are available for figure 4:

**Figure supplement 1.** Feedback-driven vs instruction-based EV in the Instructed Group.

**Figure supplement 1—source data 1.** Feedback-driven vs instruction-based EV: Instructed Group Learners (n = 20).

**Figure supplement 1—source data 2.** Neural correlates of instruction-based EV: Entire Instructed Group (n = 30).

entire ROI. Although these effects were somewhat weak when limited to learners, effects increased in magnitude when we examined the entire Instructed Group, including individuals who did not exhibit measurable SCRs (see *Figure 4—figure supplement 1* and *Figure 4—figure supplement 1—source data 2*). When we included all Instructed Group participants, we

also observed negative correlations with instruction-based EV in the VMPFC/OFC. ROI-wise analyses across the entire Instructed Group revealed that the main effect of Region (F(2, 60) = 7.12, p =0.0017) was driven by both bilateral amygdala specificity for feedback-driven EV (t(1,29) = 2.93, p =0.0066) as well as VMPFC/OFC specificity for instructed EV (t(1,29) = 2.29, p =0.0293).

## Verifying the models: Dissociable effects of instructed reversals in the instructed group

Quantitative models revealed that the striatum and VMPFC/OFC track aversive learning that updates with instructions, whereas the amygdala learns from aversive feedback irrespective of instruction. To further verify these dissociations with traditional contrast-based analyses that were independent of learning models and observed behavior, we employed a task-based, event-related General Linear Model (GLM). This allowed us to isolate the effects of instructed reversals within the Instructed Group by focusing specifically on the trials surrounding instructions, prior to subsequent reinforcement. The analysis leverages the delay between instruction delivery and US reinforcement (*Figure 5A*) and mirrors our behavioral analysis of the immediate effects of instructed reversals on SCR. We compared brain responses pre- and post-instruction by computing a CS ([previous CS+ > previous CS-]) x Phase ([Pre - Post]) interaction analysis. This provided a within-subjects comparison between instructed reversals and feedback-driven reversals. Furthermore, this analysis is entirely independent of the Uninstructed Group, and therefore allows us to ensure that the results and conclusions reported above do not simply reflect a difference in performance or learning rate between the two groups.

We first identified regions that reversed immediately upon instruction (CS x Phase interaction). Such regions should show differential activation to the previous CS+ than CS- prior to instruction, and similar activation to the new CS+ relative to the new CS- after instructions, prior to reinforcement. We observed immediate instructed reversal effects in the ventral striatum and VMPFC/OFC. The VMPFC/OFC was identified in both ROI-based and voxel-wise analyses, while the striatal activation was evident in voxel-wise analyses (see *Figure 5B*, *Figure 5—figure supplement 1*, and *Figure 5—figure supplement 1—source data 1*). Post-hoc analyses revealed that the VS showed greater activation to the current CS+, relative to the corresponding CS-, while VMPFC/OFC showed greater deactivation to the current CS+ (t(1,19) = -4.82, p=p =0.0001). Instructed reversal effects in the striatum and VMPFC/OFC were stronger and more widespread when the entire Instructed Group was included in the analysis (see *Figure 5—figure supplement 1* and *Figure 5—figure supplement 1—source data 2*). Whole-brain exploratory analyses also revealed immediate reversals with greater activation to the current CS+ in the dorsal anterior cingulate cortex (dACC), bilateral insula, thalamus, and midbrain surrounding the periaqueductal gray (PAG), and deactivation in bilateral hippocampus (see *Figure 5B*, *Figure 5—figure supplement 1*, *Figure 5—figure supplement 1—source data 1*, and *Figure 5—figure supplement 1—source data 2*).

We also identified regions that continued to show differential responses to the previous contingencies despite the fact that instructions had been delivered (main effect of CS without interaction; i.e. [previous CS+ > previous CS-] ∪ [new CS- > new CS+]). These regions presumably update with reinforcement, as our analysis incorporates all three reversals and therefore regions must show greater activation to the new CS+ prior to each subsequent instructed reversal. Voxel-wise analyses revealed that the right amygdala showed differential responses that did not reverse upon instruction (*Figure 5B*), although effects were not significant when averaged across the entire ROI. We note that we also observed a small cluster in the left putamen that did not reverse with instructions; however this region showed greater activation to the current CS- than CS+, unlike the robust ventral striatal activation observed in all of our other analyses, which reveal greater activation with higher likelihood of aversive outcomes. Results of whole-brain exploratory analyses and across the entire sample are reported in *Figure 5—figure supplement 2*, *Figure 5—figure supplement 2—source data 1* and *2*.

Finally, we tested whether these task-based effects of instructions on neural activation were related to our quantitative models and effects on behavior. We used each individual's ρ parameter (based on within-subjects model fitting; see Materials and Methods) to characterize the behavioral effects of instructions on SCR and aversive learning. We tested for correlations between this quantity and the magnitude of the instructed reversal effect (CS x Phase interaction) as well as the main effect without reversal ([previous CS+ > previous CS-] ∪ [new CS- > new CS+]), using an exploratory

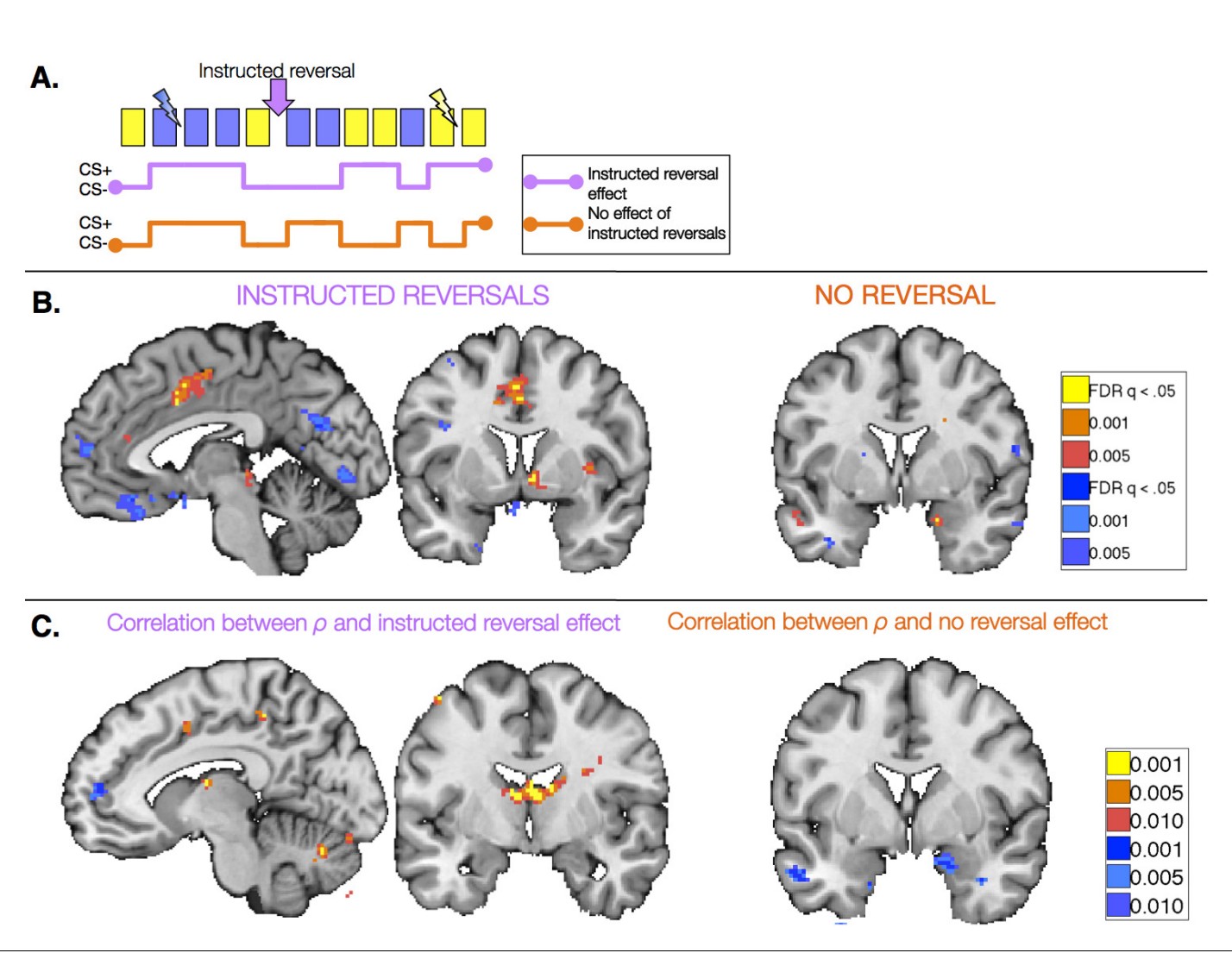

**Figure 5.** Task-based effects of instructed reversals and relationship with modeled behavior. (**A**) We examined responses in the Instructed Group surrounding the three instructed reversals to dissociate regions that are sensitive to instructions from those that learn from feedback and seem to be insensitive to instructions. Regions that are sensitive to instructions should show differential responses that reverse immediately upon instruction. The lavender timecourse depicts this pattern with greater activation on CS+ trials (blue) than CS- trials (yellow) prior to instruction, and the opposite pattern after instructions are delivered. Regions that update from aversive feedback and show no effect of instructed reversals would follow the orange timecourse, with greater activation to the previous CS+ than CS- both pre- and post-instruction. This feedback-driven pattern does not update until the new CS+ has been reinforced. (**B**) A number of regions showed differential responses that reversed upon instruction, including the right VS and VMPFC/OFC (left; see also *Figure 5—figure supplement 1*, *Figure 5—figure supplement 1—source data 1* and *2*). The VS showed greater activation to the current CS+ relative to the current CS-, whereas the VMPFC/OFC showed deactivation to the CS+. The right amygdala showed differential activation that did not reverse with instructions (right). Additional regions that did not reverse with instructions are presented in *Figure 5— figure supplement 2*, *Figure 5—figure supplement 2—source data 1* and *2*. (**C**) We conducted brain-behavior correlations to explore the relationship between neural activity in the period surrounding instructions and the magnitude of each individual's behavioral response to instructions. We tested for correlations between each individual's ρ parameter (based on within-subjects fits) and the magnitude of the reversal effect using an exploratory threshold of p<0.001, uncorrected. We observed significant correlations between ρ and the magnitude of instructed reversals in dACC (left) and the bilateral caudate tail and thalamus (right), as well as bilateral DLPFC (see *Figure 5—figure supplement 3*), suggesting that those individuals who showed stronger reversals in SCR also showed stronger reversals in these regions. In addition, we found that the individuals who showed the least evidence for updating with instructions also showed the largest non-reversing differential responses in the right amygdala (right). Full results of brain-behavior correlations are reported in *Figure 5—figure supplement 3* and *Figure 5—figure supplement 3—source data 1*.

The following source data and figure supplements are available for figure 5:

*Figure 5 continued on next page*

*Figure 5 continued*

**Figure supplement 1.** Immediate reversal with instructions (CS [previous CS+ > previous CS-] x Phase [Pre - Post] interaction) Voxelwise FDR-corrected results of regions that show immediate reversals with instructions in the Instructed Group, based on the window surrounding the delivery of instructions (see Materials and Methods).
**Figure supplement 1—source data 1.** Immediate reversal with instructions (CS x Phase interaction): Instructed Group Learners (n = 20).
**Figure supplement 1—source data 2.** Immediate reversal with instructions (CS x Phase interaction): Entire Instructed Group (n = 30).
**Figure supplement 2.** No reversal with instructions (main effect of CS without interaction; i.e.
**Figure supplement 2—source data 1.** No reversal with instructions (main effect of CS without interaction): Instructed Group Learners (n = 20).
**Figure supplement 2—source data 2.** No reversal with instructions (main effect of CS without interaction): Entire Instructed Group (n = 30).
**Figure supplement 3.** Correlations with instructed reversal (ρ) parameters.
**Figure supplement 3—source data 1.** Correlation between instructed reversal parameter (ρ) and instructed reversal effects: Instructed Group Learners (n = 20).
**Figure supplement 3—source data 2.** Correlation between instructed reversal parameter (ρ) and continued response to previous CS+ vs CS- (no reversal effect): Instructed Group Learners (n = 20).

threshold of $p < 0.001$, uncorrected. We found that the magnitude of the behavioral reversal effect was positively correlated with instructed reversal effects in bilateral dorsolateral prefrontal cortex (DLPFC), bilateral caudate tail and thalamus, dACC, and right anterior insula (see *Figure 5C*, *Figure 5—figure supplement 3*, and *Figure 5—figure supplement 3—source data 1*). Conversely, differential responses in the amygdala (which did not reverse with instructions) were strongest in those individuals whose behavior showed the weakest influence of instructions (see *Figure 5C*, *Figure 5— figure supplement 3*, and *Figure 5—figure supplement 3—source data 2*).

## Relationship between instructed reversal effects and dorsolateral prefrontal cortex response to instructions

The results reported above reveal dissociable effects of instructions on individual brain regions involved in aversive learning, and relate neural effects with observed behavior. Our final question was whether responses to instructions themselves influence subsequent learning-related neural responses. To understand how instructions influence aversive learning, we searched for brain regions that were uniquely sensitive to instructions, i.e. that showed a group difference across trials (CS onset, collapsed across CS+ and CS-). Although no regions survived FDR-correction, the left dorsolateral prefrontal cortex (DLPFC; middle frontal gyrus, peak voxel xyz = [-43 43 21]) showed greater activation across all trials in the Instructed Group at an uncorrected threshold of $p < 0.001$ (*Figure 6A*).

We then tested whether responses to instructions in this DLPFC region predicted the extent to which regions reversed immediately with instructions. We extracted DLPFC activation to the presentation of instructions for each Instructed Group participant (*Figure 6B*), and correlated this quantity with the magnitude of the instructed reversal effect (the CS x Phase interaction reported above) throughout the brain. We observed significant positive correlations in the right putamen (peak voxel xyz = [24 8 11]) and negative correlations in the VMPFC/mOFC (peak voxel xyz = [-5 40 -17]; see *Figure 6C*), although the VMPFC/mOFC cluster was slightly smaller than our cluster threshold (9 voxels rather than 10). In both cases, individuals who showed greater DLPFC activation during instruction showed stronger reversals, with greater activation to the current CS+ than CS- in the putamen, and greater deactivation to the current CS+ in the VMPFC/mOFC. Whole brain results are reported in *Figure 6—figure supplement 1* and *Figure 6—figure supplement 1—source data 1*.

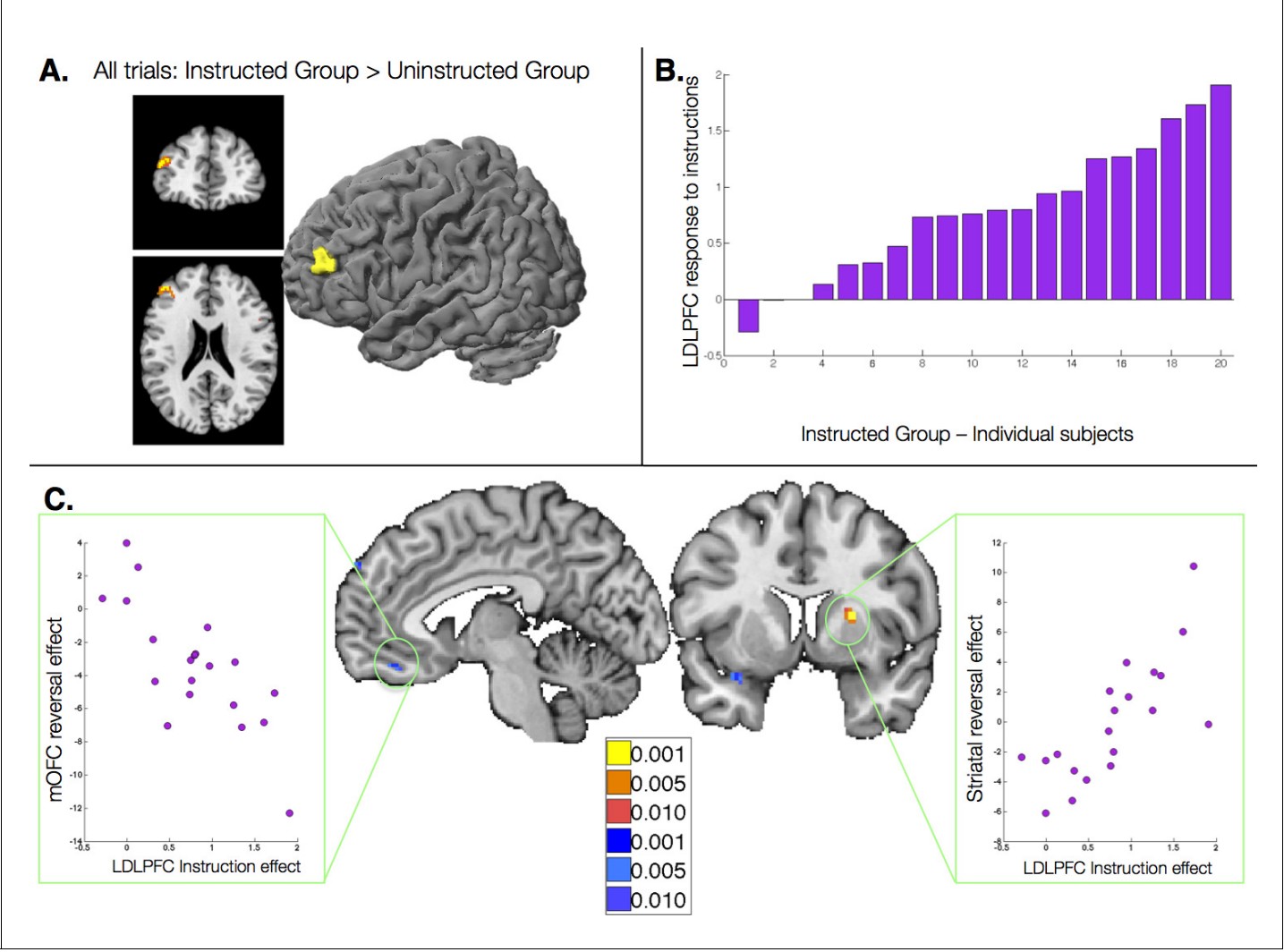

**Figure 6.** Relationship between dorsolateral prefrontal cortex response to instructions and instructed reversal effects. (**A**) The left dorsolateral prefrontal cortex (DLPFC) showed group differences across all trials, with greater activation in the Instructed Group than the Uninstructed Group. (**B**) We extracted the magnitude of the DLPFC response to instructions for each individual within the Instructed Group. **C**) The magnitude of the DLPFC response to instructions was correlated with the magnitude of instructed reversals in VMPFC/OFC (left) and dorsal putamen (right). High DLPFC responders showed larger reversals, with putamen activation to the new CS+ relative to the new CS-and VMPFC/OFC deactivation to the new CS+ relative to the new CS-. See also Figure Supplement 1 and *Figure 6—figure supplement 1—source data 1*.

The following source data and figure supplements are available for figure 6:

**Figure supplement 1.** Correlations with left dorsolateral prefrontal cortex response to instructions.

**Figure supplement 1—source data 1.** Correlation between dorsolateral prefrontal response to instructions and instructed reversal effect: Instructed Group Learners (n = 20).

## Discussion

The aim of this experiment was to determine how instructions modulate the neural mechanisms and dynamics of aversive learning, and to assess whether potentially specialized threat-detection mechanisms may be impervious to instructions, as previously hypothesized (*Ohman and Mineka, 2001*). We combined a between-groups design with experimentally manipulated instructions during aversive reversal learning, which allowed us to dissociate neural circuits that update with instructions from those that respond to reinforcement alone. Consistent with previous studies (*Wilson, 1968*;

*Grings et al., 1973*), SCRs updated immediately with instructions. Neuroimaging analyses revealed dissociable effects of instructions on regions involved in affective learning, based on both quantitative models and traditional task-based analyses. Quantitative modeling in the Uninstructed Group revealed that in the absence of instructions, the amygdala, striatum, and VMPFC/OFC tracked feedback-driven expected value (EV), consistent with a large body of literature on the circuitry that underlies reinforcement learning across species. The amygdala also followed the same feedback-driven EV timecourse in the Instructed Group, suggesting that the amygdala learned from reinforcement history, irrespective of instructions. In contrast to the amygdala, responses in the striatum and VMPFC/OFC tracked learning-related responses that updated immediately with instruction in the Instructed Group. Finally, instructed reversals in the striatum and VMPFC/OFC correlated with the DLPFC response to instructions. Below, we discuss the implications of these findings, their relationship to previous work, and important outstanding questions.

Recent studies indicate that instructions influence reward learning (*Doll et al., 2009*; *2011*; *Li et al., 2011a*). When individuals are instructed about contingencies, the prefrontal cortex influences responses in the striatum (*Doll et al., 2009*; *2011*; *Li et al., 2011a*), and VMPFC/OFC (*Li et al., 2011a*). Our results indicate that similar mechanisms drive instructed aversive learning: Striatal and VMPFC/OFC learning-related responses updated with instructions, and the strength of instructed reversals in these regions was correlated with the DLPFC response to instructions. The striatum and VMPFC/OFC make up a well-characterized network that supports error-based learning across both appetitive and aversive domains (*Balleine and O'Doherty, 2010*). The VMPFC/OFC has been linked with expected value and reversal learning (*Murray et al., 2007*; *Schiller et al., 2008*; *Rudebeck et al., 2013*), and the striatum is known to play a critical role in error-based learning (*Delgado et al., 2008b*; *Haber and Knutson, 2010*). While the striatum and VMPFC/OFC both updated with instructions, the valence of responses diverged: Striatal activation increased in anticipation of aversive outcomes, whereas the VMPFC/OFC deactivated. This is consistent with a large number of studies suggesting that the VMPFC/OFC represents a valenced outcome expectancy signal, which correlates positively with reward magnitude or the strength of appetitive outcomes (*Montague and Berns, 2002*; *Rangel and Hare, 2010*; *Grabenhorst and Rolls, 2011*; *Clithero and Rangel, 2013*), and correlates negatively with the magnitude of aversive outcomes (*Schiller et al., 2008*; *Plassmann et al., 2010*; *Atlas et al., 2014*). The signals in the striatum are also consistent with previous work showing responses predicting aversive outcomes (*Li et al., 2011b*), although we note that we found a mixture of appetitive and aversive signals here, consistent with previous work (*Seymour et al., 2005*).

Task-based analyses confirmed conclusions from quantitative approaches and revealed that differential responses in the striatum and VMPFC/OFC reversed immediately upon instruction. Instructed reversals in these regions were linked with effects of instructions on the DLPFC, consistent with findings in reward learning (*Li et al., 2011a*). The DLPFC has direct projections to the OFC (*Yeterian et al., 2012*) and rostral striatum (*Haber, 2006*), and is critical for rule maintenance, working memory, and cognitive control (*Smith and Jonides, 1999*). One model of instructed reward learning posits that the DLPFC maintains instruction information and influences striatal processing through excitatory input (*Doll et al., 2009*). Our work suggests that similar mechanisms may underlie aversive learning. Our findings are also related to previous work showing DLPFC involvement in fear conditioning in the context of contingency awareness (*Carter et al., 2006*) and cognitive regulation (*Delgado et al., 2008a*). Future studies should manipulate DLPFC (e.g. with transcranial magnetic stimulation) to test whether this region plays a causal role in the effect of instructions on associative learning in the striatum and VMPFC/OFC.

While findings in striatum, VMPFC/OFC, and DLPFC replicate and extend work on instructed reward learning, findings in the amygdala reveal an important distinction. The amygdala responded dynamically to threats in the environment similarly in both groups and showed no evidence for updating with instructions. The idea that aversive learning involves a specialized threat detection mechanism that is less sensitive to cognitive factors is consistent with suggestions concerning the amygdala's central role in automatic threat detection and salience (*Ohman and Mineka, 2001*; *Ohman, 2005*). Our findings are consistent with previous work indicating the amygdala shows differential responses during aversive learning irrespective of contingency awareness (*Tabbert et al., 2011*), and likewise that patients with unilateral temporal lobectomies exhibit differential SCRs during fear conditioning when they make expectancy ratings (*Coppens et al., 2009*). Thus, when

arousal is elicited by explicit information (e.g. through instruction or conscious expectancy), differential SCRs seem to be mediated by pathways that bypass the amygdala. Our work specifically implicates the prefrontal cortex (VMPFC, DLPFC, and DMPFC), striatum, insula, dACC, and the PAG, as these regions all updated immediately with instructions, and have been linked with skin conductance in previous work (*Boucsein, 2012*). Our results suggest that instructions can influence physiological arousal and responses in regions involved in pain and threat, consistent with previous work (*Keltner et al., 2006*; *Atlas et al., 2010*), but the amygdala seems to hold a specialized role in learning and responding as a function of aversive reinforcement history and attention-demanding stimuli in the environment.

Previous studies have attempted to isolate the role of instructions on the brain mechanisms of aversive learning. Studies of instructed fear and threat-related anxiety have examined the effects of instructions *without* reinforcement, and have shown that the amygdala can be activated by instructed knowledge (*Phelps et al., 2001*) and that left amygdala damage impairs the physiological expression of instructed fear (*Funayama et al., 2001*). Thus the amygdala might be more attuned to instructed threat in the absence of aversive reinforcement. Other studies (*Jensen et al., 2003*; *Tabbert et al., 2006*) paired instructions about contingencies with actual reinforcement, as we did here. A key distinction between our paradigm and these previous studies is that we focused on the effects of instruction in changing environments and examined dynamic learning-related responses, whereas other studies manipulated instructions in stable environments with fixed outcome probabilities, which precludes testing for dissociations in the dynamic computations that underlie instructed and feedback-driven learning.

What are the computations subserved by these different brain regions, and why might they differ in their sensitivity to instruction? We believe that the neural dissociations we identify here may reflect distinct signals with separable functions. In particular, previous work using fMRI, lesions, and electrophysiology indicates that much of the activity in VMPFC/OFC reflects the computation of expected value (*Padoa-Schioppa and Assad, 2006*; *Clithero and Rangel, 2013*). However, correlates of EV in amygdala and striatum may actually reflect two closely related quantities: associability (*Pearce and Hall, 1980*; *Behrens et al., 2007*; *Li et al., 2011a*; *Zhang et al., 2016*) and PE (*Bayer and Glimcher, 2005*; *Li et al., 2011a*; *Zhang et al., 2016*), respectively. In the current task, associability and PE each behave similarly to EV in their trial-by-trial dynamics. This is because PE reflects the difference between the EV and the (infrequent) outcomes, and because (again due to the low reinforcement rate) associability modulates up or down in the same direction as EV in response to each outcome. The differences between these signals are largely unrelated to our primary question of interest – how any of these decision variables is affected by instructions – an observation that enabled us to use a single variable to compare signaling in all three areas and test for dissociations in the effects of instructions. Thus, we chose to model the signals with a common EV variable for simplicity and transparency, and to ensure that any apparent differences between the areas were not confounded by incidental differences or artifacts of orthogonalization that arose in the modeling of three largely similar signals. However we also considered the converse concern (i.e. that apparent differences in instruction sensitivity might spuriously arise due to the fact that we are using a generic EV timeseries as a stand-in for these other quantities). We verified this was not the case by testing the hybrid model (*Pearce and Hall, 1980*; *Li et al., 2011a*; *Zhang et al., 2016*). When we model associability and PE as separate variables from EV, we see substantially the same results, with feedback-driven associability signals supported by the amygdala in both groups, and striatal PEs that update with instruction (results not shown). Finally, returning to the question of the function of these areas, an amygdala associability signal might be uniquely impermeable to instructions because of its differential function. The role of associability in models is to monitor the reliability of outcomes in the world so as, in turn, to control the learning rate for separate experiential updating about predictions of those outcomes (*Pearce and Hall, 1980*; *Behrens et al., 2007*; *Li et al., 2011a*). This function may be necessarily experiential.

The fact that amygdala-based signaling responds uniquely to threats, irrespective of instructions, might serve to prepare organisms to react to future threats in the environment. Though the vigilance of this system may benefit survival, it might also run awry in conditions characterized by persistent fear, such as anxiety and post-traumatic stress disorder. This might in turn underlie the critical role of experiential learning in successful therapy for these disorders. A better understanding of the neural

and computational underpinnings of instructed aversive learning might elucidate mediating mechanisms and/or help produce more effectively targeted interventions for these disorders.

A critical outstanding question is whether the instruction-based effects and dissociations we isolate here have parallels in other species, or whether they reflect processes that are uniquely human. While many studies show potentially homologous responses in humans, primates, and rodents, distinctions have been noted (*Wallis, 2011*). It is clear that the neural computations that support feedback-driven learning are highly conserved across species, and rely heavily on the striatum, amygdala, and VMPFC/OFC. Are the modulatory effects of verbal instructions similarly conserved? Our verbal instructions may recruit rule- and/or context-based modulatory mechanisms that operate similarly across mammalian species. For example, the strong influence of instructions on expected value coding in VMPFC/OFC may be consistent with the view that this region represents task state (*Wilson et al., 2013*), which would update when individuals are told that contingencies change. The VMPFC/OFC has been shown to guide learning when value must be inferred (*Jones et al., 2012*; *Stalnaker et al., 2014*), and inferred value can also influence learning-related signals in the dopaminergic midbrain, striatum and habenula (*Bromberg-Martin et al., 2010*; *Sadacca et al., 2016*). Likewise, rule-based processing in non-human primates is highly dependent on the prefrontal cortex (*Miller and Cohen, 2001*; *Wallis et al., 2001*), and might be linked to the instruction-based DLPFC modulation we observed here. Alternatively, it is possible that the instruction-based effects we have identified may depend on a unique human capacity for language. Future studies should combine context-based and feedback-driven reversals in other species to test for parallels with the dissociable learning systems we identify here.

A novel feature of our paradigm was the inclusion of repeated instructed reversals, which enabled us to use within-group comparisons to dissociate instructed and feedback-driven learning in the Instructed Group. We also included an Uninstructed Group as a control group to isolate purely feedback-driven knowledge – this group was not informed during initial learning or upon reversal. Future studies can use a full 2 x 2 design (instructed vs. uninstructed x initial learning vs. reversals) to determine the extent to which initial information influences subsequent learning.

## Summary

We found dissociable effects of instructions on the brain regions involved in aversive learning. Many regions, including the striatum and VMPFC/OFC, updated immediately with instructions. However, responses in the amygdala were not shaped by instructions in our task, as the amygdala required aversive feedback in order to update. These results indicate that the neural mechanisms of dynamic aversive learning include processes that perform the same computations across appetitive and aversive domains (i.e. striatal learning that updates with instructions), as well as unique patterns that may be specific to aversive learning due to evolutionary adaptations and the biological relevance of threat processing.

# Materials and methods

## Participants

Eighty right-handed English-speaking participants were enrolled in the experiment. Participants were not enrolled if they were taking any anti-depressant or anti-anxiety medication, had heart or blood pressure problems, were pregnant, or had completed a study that applied electric shocks within the previous six months. All participants provided informed consent as approved by New York University's Institutional Review Board, the University Committee on Activities Involving Human Subjects (UCAIHS; protocol #12–8965 and #13–9582). Participants were informed that the study was voluntary, and that the goals of the study were to learn about emotional responses to images that predict shock, and to learn more about how physiological responses and the activity of different regions of the brain are related to emotion and learning. Three consented participants did not participate in the experiment because we were unable to measure skin conductance. Data from eight participants were not included in the analyses either because they did not complete the study due to discomfort during the scan (n = 3) or because they slept for an extended period of time during the experiment (n = 5). Finally, intermittent signal loss prevented us from analyzing skin conductance

data for one participant, leaving a final sample of sixty-eight participants (46 Female; mean age = 22.1 years (*SE* = 0.42)).

## Stimuli and apparatus

Participants received mild electric shocks (200 ms duration, 50 pulses per second) to the right wrist using a magnetically shielded stimulator and electrodes (Grass Medical Instruments, West Warwick, RI) attached by Velcro. We measured skin conductance using shielded silver-silver chloride electrodes (BIOPAC Systems, Inc., Goleta, CA) filled with standard NaCl electrolyte gel, and attached by Velcro to the middle phalanges of the second and third fingers of the left hand. Pupillometry data were also collected, but data were corrupted for the majority of participants and are therefore not included in the current analysis.

## Experimental design

### Breath-holding test and shock calibration procedure

Upon entering the scanner suite, we affixed skin conductance electrodes to the participants' left index and middle fingers. We used a 3-second breath-hold test to ensure we could record adequate SCRs from each participant. Three participants did not show any SCR to the breath-hold, and were excluded from the experiment.

Following the breath-hold test, shock electrodes were attached to the participant's right inner wrist. Shock intensity level was calibrated using a standard work-up procedure prior to the functional portion of the experiment. Intensities started at 15V (200 ms, 50 pulses/s) and increased in 5V increments until the participant identified the maximum level he or she could tolerate without experiencing pain, with a maximum intensity of 60V. Participants were told to identify a level that was "annoying, aversive, and unpleasant but not quite painful." This intensity was recorded and used during the main experiment. The mean intensity selected across the full sample was 37.3V (*SD* = 1.1), and the two groups did not significantly differ in the voltage selected (p>0.5; Instructed Group: *M* = 36.9, *SD* = 7.84; Uninstructed Group: *M* = 37.6, *SD* = 10.3) as calibration was performed prior to random assignment. We performed the work-up procedure while the participant was in the fMRI suite to avoid context shifts between calibration and experimental portion of the study.

### Pavlovian aversive learning with reversals

During the main experiment, participants went through a Pavlovian fear conditioning procedure with a 33% reinforcement rate (see *Figure 1*) while we acquired FMRI data. Two images of angry male faces from the Ekman set (*Ekman and Oster, 1979*) were used as conditioned stimuli (CS; counterbalanced across subjects), modeled after a previous study of aversive reversal learning (*Schiller et al., 2008*). In addition, angry faces are biologically relevant (i.e. prepared) stimuli, which have been shown to be less sensitive to verbal instructions in the context of instructed extinction (*Hugdahl and Ohman, 1977*). Thus such stimuli may present a uniquely difficult test of the effects of instructions during aversive reversal learning.

Contingencies reversed every twenty trials during the task. We used two pseudorandom trial orders with the requirement that no two shocks were presented adjacent to one another, and that the same condition never repeated three times in a row. At the end of the task, participants used 5-point Likert scales to report how many reversals they had perceived (0 to >4), to rate their subjective probability of shock for each CS (0–100% in 20% increments) and how they felt about each CS (0 = very negative, 3 = neutral, 5 = very positive).

### Instructed knowledge manipulation

Individuals were randomly assigned to an Instructed Group or an Uninstructed Group. At the start of the task, participants in the Uninstructed Group saw the two CS images and were simply told "Your job is to pay attention to the relationship between the stimuli you see and the shocks that you feel." The Instructed Group was informed about shock probabilities at the start of the experiment (see *Figure 1A*) and upon contingency reversals. Reversal instructions were presented for 10 s. A fixation cross was displayed during the same reversal period for the Uninstructed Group.

Forty participants were randomly assigned to either the Instructed Group or Uninstructed Group. We analyzed the dataset at this point and determined that only a subset of participants had shown

differential skin conductance prior to the first reversal. We then collected more data through pseudorandom assignment until we had twenty 'learners' (those who showed greater SCRs, on average, to the CS+ than the CS- in the second half of the first run) in each group. Learners were defined based on mean differences in SCR to the CS+ vs. CS- in late acquisition, but we did not require statistically significant differences due to the small number of trials included in the comparison (4 trials of each condition). In total, 30 participants completed the task in the Instructed Group (20 learners), and 40 participants completed the task in the Uninstructed Group (20 learners). Two Uninstructed participants were excluded from analyses, leading to a final full sample of 68 participants. Results that include all participants are presented in Supplemental Figures and Tables, and are entirely consistent with the analyses reported in the main manuscript, which are limited to learners.

## Physiological data acquisition, processing, and analysis

Skin conductance was recorded with Acqknowledge software (BIOPAC Systems, Inc., Goleta, CA) at 200 Hz. Data were filtered using a 25-Hz low-pass FIR filter and smoothed with a Gaussian kernel of 10 samples. Trained research assistants who were blind to condition processed the trial-by-trial skin conductance data. We estimated base-to-peak amplitude in skin conductance to the first response in the 0.5- to 4.5-s latency window after CS onset. US trials were scored relative to the US presentation. Responses that were less than .02 microSiemens were considered non-responses (i.e. scores of 0). Raw SCR amplitude scores were square-root transformed to account for non-normal distributions (*Schlosberg and Stanley, 1953*) and normalized relative to the mean of the US response prior to analysis.

Data were analyzed in Matlab (Mathworks, Natick, MA, USA) using custom code for linear mixed models (available at http://wagerlab.colorado.edu/tools), and verified using the lme4 package in R (*Bates et al., 2011*). We used linear mixed models to test whether participants showed differential responses to the CS+ relative to the CS-, whether these responses reversed, and whether the magnitude of these effects differed across groups. To do this, we created a matrix for each individual (First level analyses in *Table 1*) that coded each trial as a function of Stimulus (Original CS+ versus Original CS-) and Reversal (Original contingencies versus Reversed contingencies). The Reversal regressor was coded relative to when instructions were delivered in the Instructed Group and relative to when the new context was reinforced in the Uninstructed Group (i.e. when subjects received a shock following the previous CS-). The linear mixed model evaluated trial-by-trial SCRs as a function of these two effects as well as their interaction. The Stimulus x Reversal interaction assesses the extent to which differential responses reverse, i.e. whether participants show greater SCRs to the current CS+ relative to the current CS-, irrespective of the actual stimulus. We also included a linear effect of time in the model to account for habituation, and an intercept for each subject. Group was coded at the second level, and we tested whether the slopes of the individual effects or the intercept varied across groups.

## Quantitative modeling

Our learning models assume that SCR correlates with dynamic quantities derived from feedback-driven or instructed learning models. Below we describe the quantitative models we evaluated, followed by our general procedures for model fitting.

### Feedback-driven learning model

We fit learning models to trial-by-trial SCRs from the Uninstructed Group to test whether reinforcement learning models can explain fear-conditioning behavior in the context of multiple reversals, and to generate predictors that will isolate the brain mechanisms of feedback-driven learning irrespective of instructed knowledge. The standard Rescorla-Wagner model learns an expected value (EV), denoted as for each CS, and assumes a constant learning rate ($\alpha$):

$$V_{n+1}(x_n) = V_n(x_n) + \alpha\delta_n \tag{1}$$

$$\delta_n = r_n - V_n(x_n) \tag{2}$$

(where $r = 1$ for shock, $r = 0$ for no shock, $n$ denotes the current trial, and $\delta$ = prediction error (PE), and $V = EV$). To derive the best fits for the Rescorla-Wagner model, we assumed $V_0 = 0.5$ and set $\alpha$ as

a free parameter, and evaluated regressions that assumed trial-by-trial SCRs correlate with EV (see "General procedures for model fitting and model comparison", below).

## Instruction-based learning model

To account for the influence of instructions, we introduced a new model that incorporated an additional reversal parameter ($\rho$) that determines the extent to which expected value reverses at the time when instructions are delivered. Learning proceeded as in the uninstructed learning model above until instructions were delivered (i.e. immediately following trials 20, 40, and 60). At the time of instructions, for each of the two cues ($x_a$ and $x_b$), EV was computed as the sum of the current cue's value multiplied by $1 - \rho$, plus the other cue's value multiplied by $\rho$:

$$V_{n+1}(x_a) = \rho^* V_n(x_b) + (1 - \rho)^* V_n(x_a) \tag{3}$$

$$V_{n+1}(x_b) = \rho^* V_n(x_a) + (1 - \rho)^* V_n(x_b) \tag{4}$$

Thus if $\rho = 0$, each cue retains its value, whereas if $\rho = 1$, cue $x_a$ acquires the value of cue $x_b$.

Learning then proceeded according to the feedback-driven model until the next instructions were delivered. Because the Instructed Group was informed about the original cue contingencies, we assumed asymmetrical initial expected values ($V_0(CS+) = 0.75, V_0(CS-) = 0.25$). $\rho$ was modeled as an additional free parameter.

## General procedures for model fitting and model comparison

We performed two types of model fitting, which differ in how flexibly they allow for individual differences between subjects in the model's free parameters. First, we used a mixed effects model ('Across-subjects analysis'), wherein learning parameters were modeled as fixed parameters constant within each group but we allowed slopes to vary to compensate for variability across participants (*Daw et al., 2006*). This offers a clear picture of group-level learning effects and provides clean estimates appropriate for modeling fMRI data (*Daw, 2011*). We focus on these results in the main manuscript, and parameters from these models were used to create regressors for fMRI analyses. Second, we verified that the conclusions from these aggregate fits were consistent those from with more flexible (but noisier) estimates derived from an ensemble of individual fits to each subject's data separately ('Within-subjects analysis'). Results of within-subjects analyses were entirely consistent with across-subject analyses, and are provided in *Table 2*. We used maximum likelihood estimation and Matlab's fminsearch function to determine best-fitting parameters in all models.

Our main analyses focus on fits from the across-subjects / mixed effects analysis. For each iteration of model fitting (i.e. test of a given parameter value) in Matlab's fminsearch.m program, candidate parameters were applied to each individual subject's trial order to generate a predicted timecourse of EV. For each participant, the resulting regressor was combined with an overall intercept. We used standard linear regression (through Matlab's glmfit.m function) to determine regression coefficients (based on minimizing the sum of squared errors), and then combined the regressors and the estimated coefficients to generate a predicted fit (using glmval.m). The predicted fit from each individual subject was concatenated with all other subjects to generate a group prediction, which was compared to the actual concatenated data, and aggregated residuals were used as a measure of goodness-of-fit, which were minimized over the course of ten iterations of fminsearch per model. This across-subjects approach, which estimates free parameters as fixed effects but allows slopes to vary, was used in order to avoid fixed effects model fitting (i.e. concatenating all data and treating observations as one large subject), which assumes all subjects show the same relationships with predictors and does not acknowledge that observations are nested within subjects.

To verify that the conclusions of across-subjects models held at the level of the individual, we also performed within-subjects analyses, which fit models separately for each participant. These proceeded in the same way as the mixed effects models (i.e. minimization of the sum squared error from a linear model that included an intercept and a slope for the parameters of interest) but best-fitting parameters were identified separately for each individual. We used these distributions to statistically compare the magnitude of the estimated ρ parameter across groups, and to examine brain-

behavior correlations (*Figure 5C*, *Figure 5—figure supplement 3*, *Figure 5—figure supplement 3—source data 1* and *2*).

## FMRI data acquisition and analysis

### Data acquisition

Data were acquired on a 3T Siemens Allegra head-only scanner. Functional images were acquired with a single shot gradient-echo echoplanar imaging sequence (64 x 64 matrix, TR = 2000 ms, TE = 30 ms, FOV = 192 cm, flip angle = 90°). We proscribed 36 continuous oblique angle slices (3 x 3 x 3 mm voxels) using a 30-degree head tilt relative to the plane defined by the anterior commissure-posterior commissure to maximize signal in the OFC / VMPFC (*Deichmann et al., 2003*). Minimizing dropout in this region reduced parietal cortex coverage for a number of participants; thus findings are agnostic with respect to the contribution of parietal cortex regions to aversive learning. Anatomical images were acquired with a T1-weighted protocol (256 x 256 matrix, 176 1 mm sagittal slices).

### Preprocessing

We discarded the first five volumes and performed slice time correction, coregistration, and filtering (120 Hz high-pass filter) with SPM8 (Wellcome Center Department of Imaging Neuroscience, London, UK). Functional data were smoothed with a 4mm FWHM Gaussian kernel and coregistered to anatomical data. Data were normalized to MNI space using SPM's 'avg152T1.nii' template.

### General procedures for neuroimaging analyses

We conducted four first-level whole-brain analyses using SPM8: 1) An analysis of feedback-driven aversive learning in all participants, based on EV regressors from the quantitative model fitted to SCRs from the Uninstructed Group; 2) An analysis of instructed aversive learning in the Instructed Group, based on EV regressors from the instructed learning model fitted to SCRs from the Instructed Group; 3) a direct comparison between feedback-driven and instruction-driven EV within the Instructed Group (described in more detail below) and 4) An analysis of the immediate effects of instructed reversals in the Instructed Group. First-level analyses employed the general linear model (GLM) in SPM8 without default implicit thresholding. We modeled cue onset and offset, and each EV regressor was included as a parametric modulator of the CS onset event. All events were convolved with a canonical gamma-variate hemodynamic response function (HRF). To evaluate the direct comparison between feedback-driven and instruction-driven EV within the Instructed Group, we disabled SPM's orthogonalization (in the functions spm_get_ons.m and spm_fMRI_design.m). This removes any shared variance between the two EV regressors without prioritizing one regressor over the other, which would otherwise be standard in SPM's default GLM approach when multiple regressors are included. In this analysis, we computed a second-level contrast within subjects to directly compare feedback-driven and instruction-driven EV, and contrast estimates were carried to the group level.

Group results were obtained using robust regression, which minimizes the influence of outliers (*Wager et al., 2005*). Matlab-based code is freely available at http://wagerlab.colorado.edu/tools. We also performed ROI-wise t-tests and ANOVAs using standard functions (ttest.m, ttest2.m, and anovan.m) in Matlab's Statistics and Machine Learning Toolbox. ANOVAs focusing on the feedback-driven learning model assess main effects of Group (Instructed vs. Uninstructed) and Laterality. ANOVAs focusing on the Instructed Group assess main effects of Model (instructed learning vs. feedback-driven learning) and Laterality.

In the main manuscript, we focus on results in our *a priori* ROIs: amygdala, striatum, and VMPFC/OFC (see *Figure 3—figure supplement 3*). Amygdala ROIs were defined based on the MNI template, and are available at http://wagerlab.colorado.edu/tools. Striatum ROIs were acquired from the Automated Anatomical Labeling atlas for SPM8 (http://www.gin.cnrs.fr/AAL, [*Tzourio-Mazoyer et al., 2002*]), and created by combining putamen and caudate regions to single striatal ROIs. The VMPFC/OFC ROI was functionally defined, based on the deactivation to shock in the entire sample (peak voxel xyz = [-6 38 -16]). We chose to functionally define the VMPFC/OFC because we optimized imaging parameters to recover signal in this region, and thus the center of activation lied inferior to masks based on previous data (*Schiller and Delgado, 2010*). However,

analyses that used this more superior ROI were consistent with the VMPFC/OFC effects reported here.

We used custom Matlab code (available at http://wagerlab.colorado.edu/tools) to extract and average across ROI-wise data, and we report ROI-wise analyses at standard p<0.05. We used false discovery rate (FDR) to correct for multiple comparisons in whole-brain voxel-wise analyses. For all main effects, activation clusters were defined as FDR-corrected voxels (q< 0.05) contiguous with voxels at uncorrected p<0.001 and p<0.01. Figures in the main manuscript display a maximum voxel-wise thresholds p<0.005 for precise localization. Correlations with individual difference measures (i.e. brain-behavior correlations with ρ parameters; connectivity with DLPFC), which are more exploratory, are reported at p<0.001, uncorrected (clusters required to have contiguous voxels at p<0.005 and p<0.01). For all voxelwise analyses, we imposed a minimum cluster extent of 10 voxels. We report coordinates in Montreal Neurological Institute (MNI) space. Anatomical labels were based on the SPM anatomy toolbox (*Eickhoff et al., 2005*) and verified using the Mai Atlas (*Mai et al., 2016*).

## Neural correlates of feedback-driven and instructed aversive learning

Regressors for fMRI analyses of learning-related signals were derived from the across-subjects fits to SCR, which generated best-fitting parameters. Our design had two trial orders, so we applied the best-fitting parameters to each sequence of stimuli to generate EV regressors for that were specific to each trial order. We used parameters fit across the Uninstructed Group to generate regressors for each trial order that were used to identify neural correlates of feedback-driven EV in all participants. Thus, the feedback-driven learning analyses use identical regressors for both the Uninstructed Group and Instructed Group, varying only as a function of trial order. Analyses focusing on the neural correlates of instructed aversive learning were based on parameters from the instructed learning model fit across participants in the Instructed Group.

Learning model-based neuroimaging analyses modeled cue onset and cue offset as two discrete events. EV regressors were included as parametric modulators of cue onset. Shock occurrence (0 for trials with no shock, 1 for shock trials) and PE were modeled as parametric modulators of cue offset, but are not discussed in this paper due to algebraic collinearity with EV.

### Instructed reversal analysis

We analyzed the effects of instructed reversals by examining the trials that occurred in the window between the delivery of instructions and the first time the new CS+ was reinforced with a shock (see *Figure 5*). We compared responses on these trials (new CS+ and new CS-) with responses with the same number of trials of each condition immediately prior to each reversal (previous CS+ and previous CS-). Thus we assessed a Stimulus ([Previous CS+ - Previous CS-]) x Reversal Phase ([Post – Pre]) interaction analysis. Our analysis collapsed across all three reversals at the first level. Thus regions that were identified as showing greater activation to the previous CS+ than CS- across pre- and post-reversal (i.e. Main Effect without interaction) were required to have updated over the course of the run (i.e. with reinforcement), as they had to show greater activation to the opposite contingencies by the time the next reversal was modeled. This model also coded the instruction events at the first level (10-second boxcar at the time of instructions). Beta estimates from this instruction period estimate were extracted from the DLPFC cluster identified in a between-groups comparison across all CS onset trials and correlated with the magnitude of instructed reversals in a whole-brain search.

## Acknowledgements

We would like to thank Eyal Bar-David, Rachel Mojdehbakhsh, and Keith Sanzenbach for assistance with fMRI scanning, Manurhadi Perera, Augustus Baker, and Erin Furlong for assistance with SCR data processing and subject recruitment, and Tor Wager and Vincent Costa for helpful discussion. This work was supported by an NIH grant awarded to E A. Phelps (RO1MH097085), and by the Intramural Research program of the NIH's National Center for Complementary and Integrative Health.

# Additional information

## Funding

| Funder | Grant reference number | Author |
|--------|------------------------|--------|
| National Center for Complementary and Integrative Health | | Lauren Y Atlas |
| National Institute of Mental Health | RO1MH097085 | Elizabeth A Phelps |

The funders had no role in study design, data collection and interpretation, or the decision to submit the work for publication.

## Author contributions

LYA, Conceived of and designed the experiment, Collected and analyzed the data, Wrote the manuscript; BBD, NDD, EAP, Contributed to the conception and design of the study, Interpretation of analysis and results, Writing the paper; JL, Contributed to the conception and design of the study, Writing the paper

## Author ORCIDs

Lauren Y Atlas, http://orcid.org/0000-0001-5693-4169
Nathaniel D Daw, http://orcid.org/0000-0001-5029-1430

## Ethics

Human subjects: Informed consent was obtained from all subjects. Research was approved by New York University's Institutional Review Board.

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
