## [Decision Letter]

Thank you for submitting your article "Instructed knowledge shapes feedback-driven aversive learning in striatum and orbitofrontal cortex, but not the amygdala" for consideration by *eLife*. Your article has been reviewed by Geoffrey Schoenbaum and Ben Seymour, and the evaluation has been overseen by Timothy Behrens as Reviewing and Senior Editor.

The reviewers have discussed the reviews with one another and the Reviewing Editor has drafted this decision to help you prepare a revised submission.

This study highlights the complexities of human learning in an elegant fashion. It shows that learning from instructions can be dissociated from learning from experience both behaviourally and neurally. When the two learning-types conflict, the brain simultaneously contains estimates from both types of learning in different structures. The ventral prefrontal cortex (OFC, vmPFC) reflect the explicit knowledge derived from instructions, and the amygdala circuitry reflects the experientially derived estimates. The separate learning systems are shown both in a comparison between groups learning with and without explicit instructions, and demonstrated to exist in parallel within the instructed group. They appear to be strong and robust.

This separation is important for our understanding of human learning and how it relates to animal learning. It is important for our understanding of the interactions between these highly connected neuroanatomical structures. Because the result is found in the context of fear conditioning, it is also clinically informative for potential strategies for alleviating disorders of fear and anxiety.

The BRE and the two reviewers were universally positive about the interest of the study. For example:

In this study the authors compared instructed versus non-instructed learning in a Pavlovian conditioning task in which cues were paired with mild shock. Models including a parameter to account for changes with instruction were fitted to SCR data. The results showed that instruction resulted in a remarkable improvement anticipating shock appropriately after the correct cue, both immediately after reversal and across the block. Both groups also showed a slower, presumably reinforcement mediated change in anticipating shock. The latter correlated best with signal in amygdala while the former was best related to activity in orbitofrontal cortex. Overall the study is brilliant, presenting a simple straightforward approach to the critical question of how higher order knowledge impacts underlying reinforcement learning mechanisms. The results are enticing because they fit well with some of our preconceived notions of how the systems may work but they also I think suggest some provocative overlap in the two theoretically dissociated systems. I especially like the identification of the amygdala as not responsive as I think this fits well with other data. I have a couple questions and suggestions which I think reflect less issues than places where I think the authors might extend or broaden the impact of their study, either by making more clear some of the more subtle results or by applying their idea more broadly.

This interesting study aims to identify the neural correlates of the effect on instruction on Pavlovian aversive learning, using a reversal task in humans with SCR and BOLD measurement. The main behavioural finding is that SCRs are sensitive to instruction, and the main brain imaging finding is that striatal and mPFC BOLD responses are sensitive to instruction, but amygdala is not. On the whole, the analyses are correct and data robust.

The reviewers did, however, have some queries. These related to the modelling choices, and to some clarifications. These corrections will not require re-review except by the BRE member (TB; Timothy Behrens). Indeed, all the suggested changes are really optional.

Questions relating to the modelling choices (see note from TB below).

My main concern relates to the model. The authors have previously shown that uninstructed reversal learning is best captured by a dynamic learning rate model – the 'hybrid TD model (Li et al., Doll et al., based on Sutton's gain adaption model) – but there's no mention of any of these studies here. In these studies, the SCR and amygdala is shown to correlate with the 'associability'. This model specifically captures the dynamic nature of learning in reversal, so it appears very surprising that the authors not test or even mention it.

In a similar vein, they model only expected value signals, and not prediction errors, which has been repeatedly demonstrated (including by the authors) to best explain striatal responses. So again, why did they not model prediction errors here?

It may be that the authors are trying to present the simplest analysis, but some explanation is required.

Note from TB. The above reviewer concerns obviously relate to changes you have made since initial submission in response to our triage comments about the possibility of establishing clean double-dissociations with the two different variables in the associability model. After discussions post-review, the reviewer is happy with this. We all agree, however, that it would be useful to explain this to the reader.

Looking at the data in more detail, the effect of reversal in the uninstructed model is weak (basic post-reversal contrasts are non-significant, but this is in itself not a problem as they don't capture learning dynamics. Even so, it's not a problem for the contrasts). I guess this because the contingency is low (33%). Perhaps this is worth a comment to alleviate potential reader concerns.

Questions about interpretation:

As I noted, I am impressed (and fond of) the dissociation between the amygdala and orbital signals and correlations with the different types of learning. However, I was a bit confused by the strong claims for this dissociation in the Abstract, Introduction and Discussion, when the data shows that both areas and striatum are engaged in the non-instruction group. I may not fully grasp the complexities of the analyses, but I wonder if this suggests an involvement of the prefrontal areas generally in reinforcement learning, and that the effects of instruction modulate this. I think this is a slightly different conclusion than I took away from the current focus on dissociation. I wonder also if clarifying this might help me understand the other thing I found confusing, which was the inverse relationship between instructed behavior and prefrontal signal. If I understood correctly, in the instructional group, there was a negative correlation? If this is driving the switch in SCR/EV, why is less signal associated with more change?

The second place I would suggest the authors consider changes or clarification is in their definition of instructed learning. This has an intuitively obvious meaning to them, since they test humans. Instructed learning is learning driven by the verbal or written advice given to the subjects prior to testing in their design clearly. Yet I wonder if they mean to restrict their conclusions this way? It reminds me a bit of the original definition of declarative memory used to define hippocampal function, where a focus on language initially introduced a distinction in function across species that seems unnecessary in retrospect. If the authors really mean their results to be uniquely human, then this is their prerogative in my opinion. But I think this may not be the case. Specifically, to the extent that instructions are simply a mechanism by which knowledge not directly acquired through experience can influence behavior/learning/neural activity, then I think there is ample evidence that this can happen in other species. For example, single unit activity has been shown to change under conditions somewhat similar to what is demonstrated here: Stalnaker et al., Nature Communications 5:3926; Bromberg-Martin et al. Journal of Neurophysiology 104:1068-1076. And such knowledge has been directly demonstrated to modulate learning: Jones J et al. Science 338:953-956. These are just a couple examples. Obviously there must be unique systems involved very specifically representing/interpreting instructions that are language based. But I think the current approach does not necessarily isolate that contrast.

Note here from TB (who agrees with this point) – On this point, I guess it would be interesting to note how this relates to the ideas of state-spaces etc. that have been emerging in the ventral prefrontal literature. As the reviewer (GS) notes, I think there are people who would believe that explicit instruction (language) is one particularly efficient way to establish such a representation, but not the only way. Interested to hear your views if you think it appropriate.

---

## [Author Response]

*[…] The reviewers did, however, have some queries. These related to the modelling choices, and to some clarifications. These corrections will not require re-review except by the BRE member (TB). Indeed, all the suggested changes are really optional.*

*Questions relating to the modelling choices (see note from TB below).*

*My main concern relates to the model. The authors have previously shown that uninstructed reversal learning is best captured by a dynamic learning rate model – the 'hybrid TD model (Li et al., Doll et al., based on Sutton's gain adaption model) – but there's no mention of any of these studies here. In these studies, the SCR and amygdala is shown to correlate with the 'associability'. This model specifically captures the dynamic nature of learning in reversal, so it appears very surprising that the authors not test or even mention it.*

*In a similar vein, they model only expected value signals, and not prediction errors, which has been repeatedly demonstrated (including by the authors) to best explain striatal responses. So again, why did they not model prediction errors here?*

*It may be that the authors are trying to present the simplest analysis, but some explanation is required.*

*Note from TB. The above reviewer concerns obviously relate to changes you have made since initial submission in response to our triage comments about the possibility of establishing clean double-dissociations with the two different variables in the associability model. After discussions post-review, the reviewer is happy with this. We all agree, however, that it would be useful to explain this to the reader.*

We appreciate the suggestion that we explain the relationship between these models for the reader, and we do so now in the Discussion. As the BRE and Senior Review Board pointed out in our original submission, using a simplified model with a single quantity allows us to test for double dissociations.

We have indeed evaluated the hybrid model, modified to capture the influence of instructions. We find that amygdala-dependent associability tracks feedback alone, whereas striatal prediction error updates with instruction (i.e. we see significant interactions between group and source of learning in ventral striatal PE, with Instructed PE only in the Instructed Group, and feedback-driven PE only in the Uninstructed Group). These results are entirely consistent with the conclusions we make here, as both models reveal that instructions update learning in the striatum and orbitofrontal cortex, but not the amygdala. However, the current model allows for direct tests of dissociations. We now discuss the fact that expected value reflects the joint combination of learning rate and prediction error, and that dynamic learning models may provide fine-grained tests of the precise computations that underlie the dissociations we see here, and we summarize findings from this alternative approach and explain the relationship with the more parsimonious model we report here (Discussion).

Finally, the reviewer asked why we did not model prediction errors in the current analysis, given our focus on responses in the striatum. When we jointly model value, prediction error, and shock, coefficients for PE may be unreliable due to algebraic collinearity in the basic Rescorla-Wagner model, as we now acknowledge in the first paragraph of the subsection “Feedback-driven learning is modulated by instructions”. However, the hybrid model decouples these quantities due to the time-varying learning rate. As noted above, when we compare instructed and feedback-driven variants of the hybrid model, we observe feedback-driven PEs in the Uninstructed Group, and these update with instructions in the Instructed Group. We now discuss these findings in the Discussion, although we opted not to include these results in the manuscript in the interest of simplicity.

*Looking at the data in more detail, the effect of reversal in the uninstructed model is weak (basic post-reversal contrasts are non-significant, but this is in itself not a problem as they don't capture learning dynamics. Even so, it's not a problem for the contrasts). I guess this because the contingency is low (33%). Perhaps this is worth a comment to alleviate potential reader concerns.*

The reviewer is correct that reversals were slow in the Uninstructed Group, although we do see significant reversals if we examine the second half of each block, which is indeed consistent with the low reinforcement rate, as the reviewer has noted. We now report this explicitly in the subsection “Instructions influence skin conductance responses”. The reviewer points out that this weak behavior also leads to minimal impact of reversals in our feedback-driven model, although we do note that there are reversals evident in Figure 2, and that these effects may be magnified dependent on the coefficient when the model is fit to behavior. Reversals may be somewhat better captured by the hybrid model, as this allows learning rates to vary dynamically and associability increases when contingencies change. Since results of the hybrid model were highly consistent with the results of the simplified model we present here, we do not feel that the slow reversals substantively influenced our conclusions.

*Questions about interpretation:*

*As I noted, I am impressed (and fond of) the dissociation between the amygdala and orbital signals and correlations with the different types of learning. However, I was a bit confused by the strong claims for this dissociation in the Abstract, Introduction and Discussion, when the data shows that both areas and striatum are engaged in the non-instruction group. I may not fully grasp the complexities of the analyses, but I wonder if this suggests an involvement of the prefrontal areas generally in reinforcement learning, and that the effects of instruction modulate this. I think this is a slightly different conclusion than I took away from the current focus on dissociation.*

We appreciate the reviewer’s remarks regarding the dissociation. We agree that our work in the Uninstructed Group is consistent with a large body of work demonstrating experiential reinforcement learning signals in the amygdala, striatum, and orbitofrontal cortex in both human and animal models. We interpret the reviewer’s statement as concern regarding the notion of [double] dissociation. Indeed, we do not mean to claim that we identify regions that learn *only* from instruction. Instead we mean that there is a single dissociation between regions that update with instruction (orbitofrontal cortex and striatum) and those that do not (the amygdala). We have clarified the important point that in the absence of instructions, all of the regions of interest are implicated in reinforcement learning, based on results in the Uninstructed Group (Discussion, first paragraph). Evidence for the dissociation is most apparent based on our within-subjects analyses in the Instructed Group.

*I wonder also if clarifying this might help me understand the other thing I found confusing, which was the inverse relationship between instructed behavior and prefrontal signal. If I understood correctly, in the instructional group, there was a negative correlation? If this is driving the switch in SCR/EV, why is less signal associated with more change?*

We believe that the reviewer’s question concerns the inverse correlation between orbitofrontal/ventromedial cortex and expected value (e.g. Figure 3 and Figure 4), and likewise the negative correlation between dorsolateral prefrontal increases and differential responses in this region (Figure 6). Our findings are consistent with many fMRI studies of fear, pain, and aversive learning, which show deactivation in this region in anticipation of, and in response to, aversive outcomes. This can be understood in terms of a valenced value-coding function, such that the region shows increased activation in expectation of increasing rewards (positive correlation with positive expected value), and likewise, is negatively correlated with the magnitude of aversive outcomes. We have clarified the valenced BOLD signals and relationship to the literature on aversive learning in the second paragraph of the Discussion.

*The second place I would suggest the authors consider changes or clarification is in their definition of instructed learning. This has an intuitively obvious meaning to them, since they test humans. Instructed learning is learning driven by the verbal or written advice given to the subjects prior to testing in their design clearly. Yet I wonder if they mean to restrict their conclusions this way? It reminds me a bit of the original definition of declarative memory used to define hippocampal function, where a focus on language initially introduced a distinction in function across species that seems unnecessary in retrospect. If the authors really mean their results to be uniquely human, then this is their prerogative in my opinion. But I think this may not be the case. Specifically, to the extent that instructions are simply a mechanism by which knowledge not directly acquired through experience can influence behavior/learning/neural activity, then I think there is ample evidence that this can happen in other species. For example, single unit activity has been shown to change under conditions somewhat similar to what is demonstrated here: Stalnaker et al., Nature Communications 5:3926; Bromberg-Martin et al. Journal of Neurophysiology 104:1068-1076. And such knowledge has been directly demonstrated to modulate learning: Jones J et al. Science 338:953-956. These are just a couple examples. Obviously there must be unique systems involved very specifically representing/interpreting instructions that are language based. But I think the current approach does not necessarily isolate that contrast.*

*Note here from TB (who agrees with this point) – On this point, I guess it would be interesting to note how this relates to the ideas of state-spaces etc. that have been emerging in the ventral prefrontal literature. As the reviewer (GS) notes, I think there are people who would believe that explicit instruction (language) is one particularly efficient way to establish such a representation, but not the only way. Interested to hear your views if you think it appropriate.*

We thank the reviewer and BRE for encouraging us to address the possibility that our effects are not unique to humans, or dependent on language. We are in strong agreement, and indeed we do believe that the dissociations we isolate here may reflect the broader function of context or rule-based modulation that is unlikely to be uniquely human, or language-dependent. We now expand our discussion of potential homologues in other mammalian species (including Wilson et al.’s state space account and other related work) in the eighth paragraph of the Discussion. Of course, we feel that it would be premature to assert that our findings reflect a general mechanism conserved across species on the basis of the single study presented here, but we now explicitly acknowledge the question of whether these mechanisms translate across species in the Abstract, Introduction (first paragraph) and Discussion. We hope this expanded discussion will encourage researchers working on related questions in non-human primates and/or other species, as we are quite excited about the possibility of developing related paradigms to test for parallel dissociations in animal models, and we do think the literature is promising with respect to this possibility.